# Oracle Complexity in Nonsmooth Nonconvex Optimization

**Guy Kornowski**
Weizmann Institute of Science
`guy.kornowski@weizmann.ac.il`

**Ohad Shamir**
Weizmann Institute of Science
`ohad.shamir@weizmann.ac.il`

## Abstract

It is well-known that given a smooth, bounded-from-below, and possibly non-convex function, standard gradient-based methods can find $\epsilon$-stationary points (with gradient norm less than $\epsilon$) in $\mathcal{O}(1/\epsilon^2)$ iterations. However, many important nonconvex optimization problems, such as those associated with training modern neural networks, are inherently not smooth, making these results inapplicable. In this paper, we study nonsmooth nonconvex optimization from an oracle complexity viewpoint, where the algorithm is assumed to be given access only to local information about the function at various points. We provide two main results (under mild assumptions): First, we consider the problem of getting *near* $\epsilon$-stationary points. This is perhaps the most natural relaxation of *finding* $\epsilon$-stationary points, which is impossible in the nonsmooth nonconvex case. We prove that this relaxed goal cannot be achieved efficiently, for any distance and $\epsilon$ smaller than some constants. Our second result deals with the possibility of tackling nonsmooth nonconvex optimization by reduction to smooth optimization: Namely, applying smooth optimization methods on a smooth approximation of the objective function. For this approach, we prove an inherent trade-off between oracle complexity and smoothness: On the one hand, smoothing a nonsmooth nonconvex function can be done very efficiently (e.g., by randomized smoothing), but with dimension-dependent factors in the smoothness parameter, which can strongly affect iteration complexity when plugging into standard smooth optimization methods. On the other hand, these dimension factors can be eliminated with suitable smoothing methods, but only by making the oracle complexity of the smoothing process exponentially large.

## 1 Introduction

We consider optimization problems associated with functions $f : \mathbb{R}^d \to \mathbb{R}$, where $f(\cdot)$ is Lipschitz continuous and bounded from below, but otherwise satisfies no special structure, such as convexity. Clearly, in high dimensions, it is generally impossible to efficiently find a global minimum of a nonconvex function. However, if we relax our goal to finding (approximate) stationary points, then the nonconvexity is no longer an issue. In particular, it is known that if $f(\cdot)$ is *smooth* – namely, differentiable and with a Lipschitz continuous gradient – then for any $\epsilon > 0$, simple gradient-based algorithms can find $\mathbf{x}$ such that $\|\nabla f(\mathbf{x})\| \leq \epsilon$, using only $\mathcal{O}(1/\epsilon^2)$ gradient computations, independent of the dimension (see for example [30, 22, 11]).

Unfortunately, many optimization problems of interest are inherently *not* smooth. For example, when training modern neural networks, involving max operations and rectified linear units, the associated optimization problem is virtually always nonconvex as well as nonsmooth. Thus, the positive results above, which crucially rely on smoothness, are inapplicable. Although there are positive results even for nonconvex nonsmooth functions, they tend to be either purely asymptotic in nature (e.g.,

35th Conference on Neural Information Processing Systems (NeurIPS 2021).

[8, 23, 36, 16, 26]), depend on the internal structure and representation of the objective function, or require additional assumptions which many problems of interest do not satisfy, such as weak convexity or some separation between nonconvex and nonsmooth components[1] (e.g., [12, 13, 18, 10, 15, 17, 6]). This leads to the interesting question of developing black-box algorithms with non-asymptotic guarantees for nonsmooth nonconvex functions, without assuming any special structure.

In this paper, we study this question via the well-known framework of oracle complexity [27]: Given a class of functions $\mathcal{F}$, we associate with each $f \in \mathcal{F}$ an *oracle*, which for any $\mathbf{x}$ in the domain of $f(\cdot)$, returns local information about $f(\cdot)$ at $\mathbf{x}$ (such as its value and gradient[2]). We consider iterative algorithms which can be described via an interaction with such an oracle: At every iteration $t$, the algorithm chooses an iterate $\mathbf{x}_t$, and receives from the oracle local information about $f(\cdot)$ at $\mathbf{x}_t$, which is then used to choose the next iteration $\mathbf{x}_{t+1}$. This framework captures essentially all iterative algorithms for black-box optimization. In this framework, we fix some iteration budget $T$, and ask what properties can be guaranteed for the iterates $\mathbf{x}_1, \ldots, \mathbf{x}_T$, as a function of $T$ and uniformly over all functions in $\mathcal{F}$ (for example, how close to optimal they are, whether they contain an approximately-stationary point, etc.). Unfortunately, as recently pointed out in [37], neither small optimization error nor small gradients can be obtained for nonsmooth nonconvex functions with such local-information algorithms: Indeed, approximately-stationary points can be easily "hidden" inside some arbitrarily small neighborhood, which cannot be found in a bounded number of iterations.

Instead, we consider here two alternative approaches to tackle nonsmooth nonconvex optimization, and provide new oracle complexity results for each. We note that Zhang et al. [37] recently proposed another promising approach, by defining a certain relaxation of approximate stationarity (so-called $(\delta, \epsilon)$-stationarity), and remarkably, prove that points satisfying this relaxed goal can be found via simple iterative algorithms with provable guarantees. However, there exist cases where their definition does not resemble a stationary point in any intuitive case, and thus it remains to be seen whether it is the most appropriate one. We further discuss the pros and cons of their approach in Appendix B.

In our first contribution, we consider relaxing the goal of finding approximately-stationary points, to that of finding *near*-approximately-stationary points: Namely, getting $\delta$-close to a point $\mathbf{x}$ with a (generalized) gradient of norm at most $\epsilon$. This is arguably the most natural way to relax the goal of finding $\epsilon$-stationary points, while hopefully still getting meaningful algorithmic guarantees. Moreover, approaching stationary points is feasible in an asymptotic sense (see for instance [17]). Unfortunately, we formally prove that this relaxation already sets the bar too high: For a very large class of gradient-based algorithms (and in fact, all of them under a mild assumption), it is impossible to find near-approximately-stationary point with worst-case finite-time guarantees, for small enough constant $\delta, \epsilon$.

In our second contribution, we consider tackling nonsmooth nonconvex optimization by reduction to smooth optimization: Given the target function $f(\cdot)$, we first find a smooth function $\tilde{f}(\cdot)$ (with Lipschitz gradients) which uniformly approximates it up to some arbitrarily small parameter $\epsilon$, and then apply a smooth optimization method on $\tilde{f}(\cdot)$. Such reductions are common in convex optimization (e.g., [28, 7, 1]), and intuitively, should usually lead to points with meaningful properties with respect to the original function $f(\cdot)$, at least when $\epsilon$ is small enough. For example, it is known that stationary points of $f(\cdot)$ are the limit of approximately-stationary points of appropriate smoothings of $f(\cdot)$, as $\epsilon \to 0$ [32, Thm. 9.67].

This naturally leads to the question of how we can find a smooth approximation of a Lipschitz function $f(\cdot)$. Inspecting existing approaches for smoothing nonconvex functions, we notice an interesting trade-off between computational efficiency and the smoothness of the approximating function: On the one hand, there exist optimization-based methods from the functional analysis literature (in particular, Lasry-Lyons regularization [25]) which yield essentially optimal gradient Lipschitz parameters, but are not computationally efficient. On the other hand, there exist simple, computationally tractable methods (such as randomized smoothing [19]), which unfortunately lead to much worse gradient Lipschitz parameters, with strong scaling in the input dimension. This in turn leads to larger iteration complexity, when plugging into standard smooth optimization methods. Is this kind of trade-off between computational efficiency and smoothness necessary? Considering

---

[1]A trivial example arising in deep learning, which does not satisfy most such structural assumptions, is the negative of the ReLU function, $x \mapsto -\max\{0, x\}$.

[2]For non-differentiable functions, we use a standard generalization of gradients following [14], see Sec. 2 for details.

this question from an oracle complexity viewpoint, we prove that this trade-off is indeed necessary under mild assumptions: If we want a smoothing method whose oracle complexity is polynomial in the problem parameters, we must accept that the gradient Lipschitz parameter may be no better than that obtained with randomized smoothing (up to logarithmic factors). Thus, in a sense, randomized smoothing is an optimal nonconvex smoothing method among computationally efficient ones.

Overall, we hope that our work motivates additional research into black-box algorithms for nonconvex, nonsmooth optimization problems, with rigorous finite-time guarantees.

Our paper is structured as follows. In Sec. 2, we formally introduce the notations and terminology that we use. In Sec. 3, we present our results for getting near approximately stationary points. In Sec. 4, we present our results for smoothing nonsmooth nonconvex functions. We conclude in Sec. 5 with a discussion of open questions. The supplementary material contains, besides the full proofs, a further discussion of the notion of $(\delta, \epsilon)$-stationarity from [37] (Appendix B), and a proof that the dimension-dependencies arising from randomized smoothing provably affects the iteration complexity of vanilla gradient descent (Appendix C).

## 2  Preliminaries

**Notation.** We let bold-face letters (e.g., $\mathbf{x}$) denote vectors. $\mathbf{0}$ is the zero vector in $\mathbb{R}^d$ (where $d$ is clear from context), and $\mathbf{e}_1, \mathbf{e}_2, \ldots$ are the standard basis vectors. Given a vector $\mathbf{x}$, $x_i$ denotes its $i$-th coordinate, and $\bar{\mathbf{x}}$ denotes the normalized vector $\mathbf{x}/\|\mathbf{x}\|$ (assuming $\mathbf{x} \neq \mathbf{0}$). $\langle \cdot, \cdot \rangle$, $\| \cdot \|$ denote the standard Euclidean dot product and its induced norm over $\mathbb{R}^d$, respectively. For any real number $x$, we denote $[x]_+ := \max\{x, 0\}$. Given two functions $f(\cdot), g(\cdot)$ on the same domain $\mathcal{X}$, we define $\|f - g\|_\infty = \sup_{\mathbf{x} \in \mathcal{X}} |f(\mathbf{x}) - g(\mathbf{x})|$. We denote by $\mathcal{S}^{d-1} := \{\mathbf{x} \in \mathbb{R}^d \mid \|\mathbf{x}\| = 1\}$ the unit sphere. For any natural $N$, we abbreviate $[N] := \{1, \ldots, N\}$. We occasionally use standard big-O asymptotic notation, with $\mathcal{O}(\cdot), \Theta(\cdot), \Omega(\cdot)$ hiding constants, $\tilde{\mathcal{O}}(\cdot), \tilde{\Theta}(\cdot), \tilde{\Omega}(\cdot)$ hiding constants and logarithmic factors, and $\text{poly}(\cdot)$ meaning polynomial factors.

**Gradients and generalized gradients.** If a function $f : \mathbb{R}^d \to \mathbb{R}$ is differentiable at $\mathbf{x}$, we denote its gradient by $\nabla f(\mathbf{x})$. For possibly non-differentiable functions, we let $\partial f(\mathbf{x})$ denote the set of *generalized* gradients (following [14]), which is perhaps the most standard extension of gradients to nonsmooth nonconvex functions. For Lipschitz functions (which are almost everywhere differentiable by Rademacher's theorem), one simple way to define it is

$$\partial f(\mathbf{x}) := \text{conv}\{\mathbf{u} : \mathbf{u} = \lim_{k \to \infty} \nabla f(\mathbf{x}_k), \mathbf{x}_k \to \mathbf{x}\}$$

(namely, the convex hull of all limit points of $\nabla f(\mathbf{x}_k)$, over all sequences $\mathbf{x}_1, \mathbf{x}_2, \ldots$ of differentiable points of $f(\cdot)$ which converge to $\mathbf{x}$). With this definition, a *stationary point* with respect to $f(\cdot)$ is a point $\mathbf{x}$ satisfying $\mathbf{0} \in \partial f(\mathbf{x})$. Also, given some $\epsilon \geq 0$, we say that $\mathbf{x}$ is an $\epsilon$-*stationary point* with respect to $f(\cdot)$, if there is some $\mathbf{u} \in \partial f(\mathbf{x})$ such that $\|\mathbf{u}\| \leq \epsilon$.

**Oracles.** In this paper, we consider two types of oracles:

- A *first-order* oracle $\mathsf{O}_f^1(\mathbf{x})$, which given a function $f(\cdot)$ and a point $\mathbf{x}$, returns $f(\mathbf{x})$ and $\partial f(\mathbf{x})$. We note that this differs somewhat from the standard formulation of first-order oracles in the literature, where it is assumed that at points of non-differentiability, the oracle only returns some element in the generalized gradient set $\partial f(\mathbf{x})$. However, this means that our oracle complexity lower bounds hold under weaker assumptions, since we only provide more information to the algorithm. We use this type of oracle in the results of Sec. 3, but discuss possible extensions in Remark 2.

- An *all-derivatives* oracle $\mathsf{O}_f^\infty(\mathbf{x})$, which returns $f(\mathbf{x}), \partial f(\mathbf{x})$, as well as all higher-order derivatives of $f(\cdot)$ at $\mathbf{x}$, if they exist. This oracle provides more information than a first-order oracle, so any lower bound with respect to an all-derivatives oracle trivially applies with respect to a first-order oracle. Some of our bounds apply with respect to such a powerful oracle, and we present them in this generality, although we note that in the nonconvex-nonsmooth setting, it remains to be seen whether there is any use for higher-order derivative information. We use this type of oracle in the results of Sec. 4.

# 3 Hardness of Getting Near Approximately-Stationary Points

In this section, we present our first main result, which establishes the hardness of getting near approximately-stationary points.

To avoid trivialities, and following [37], we will focus on functions $f(\cdot)$ which are Lipschitz and bounded from below. In particular, we will assume that $f(\mathbf{0}) - \inf_{\mathbf{x}} f(\mathbf{x})$ is upper bounded by a constant. We note that this is without loss of generality, as $\mathbf{0}$ can be replaced by any other fixed reference point. Moreover, we will focus on two broad families of algorithms, which together span nearly all algorithms of interest: The first is the class of all *deterministic* algorithms (denoted as $\mathcal{A}_{det}$), which are all oracle-based algorithms where $\mathbf{x}_1$ is chosen deterministically, and for all $t > 1$, $\mathbf{x}_t$ is some deterministic function of $\mathbf{x}_1$ and the previously observed values and gradients. The second is the class of all *zero-respecting* algorithms (denoted as $\mathcal{A}_{zr}$), which are all deterministic or randomized oracle-based algorithms whose sequence of query points $\mathbf{x}_1, \mathbf{x}_2, \ldots$ satisfy $\text{support}\{\mathbf{x}_t\} \subseteq \bigcup_{i<t} \text{support}(\partial f(\mathbf{x}_i))$ for all $t$, where $\text{support}(M)$ (for some $M \subseteq \mathbb{R}^d$) is defined as $\{i \in [d] : \exists \mathbf{y} \in M : y_i \neq 0\}$: Namely, the set of coordinates on which $M$ has some elements with non-zero values. The notion of zero-respecting algorithms was introduced in [11], and generalizes the common notion of linear-span algorithms [31], where $\mathbf{x}_t$ is assumed to lie in the span of previous gradients. The class of zero-respecting algorithms includes nearly all algorithms of practical interest in our setting, including some which are not linear-span algorithms, such as coordinate descent methods [29]. Our main result in this section is the following:

**Theorem 1.** *There exist a large enough absolute constant $C > 0$ and a small enough absolute constant $c > 0$ such that the following holds: For any algorithm in $\mathcal{A}_{det} \cup \mathcal{A}_{zr}$ interacting with a first-order oracle, any $T > 1$, and any $d \geq 2T$, there is a function $f(\cdot)$ on $\mathbb{R}^d$ such that*

- $f(\cdot)$ *is $C$-Lipschitz, and $f(\mathbf{0}) - \inf_{\mathbf{x}} f(\mathbf{x}) \leq C$.*

- *With probability at least $1 - T\exp(-cd)$ over the algorithm's randomness (or deterministically if the algorithm is deterministic), the iterates $\mathbf{x}_1, \ldots, \mathbf{x}_T$ produced by the algorithm satisfy $\inf_{\mathbf{x} \in \mathcal{S}} \min_{t \in \{1,\ldots,T\}} \|\mathbf{x}_t - \mathbf{x}\| \geq c$, where $\mathcal{S}$ is the set of $c$-stationary points of $f(\cdot)$.*

The theorem implies that for a very large family of algorithms, it is impossible to obtain worst-case guarantees for finding near-approximately-stationary points of Lipschitz, bounded-from-below functions unless $T$ has exponential dependence on $d$.

Before continuing, we note that the result can be extended to *any* oracle-based algorithm (randomized or deterministic) under a widely-believed assumption – see remark 5 in the proof for details.

**Remark 1** (More assumptions on $f(\cdot)$). *The Lipschitz functions $f(\cdot)$ used to prove the theorem are based on a simple composition of affine functions, the Euclidean norm function $\mathbf{x} \mapsto \|\mathbf{x}\|$, and the max function. Thus, the result also holds for more specific families of functions considered in the literature, which satisfy additional regularity properties, as long as they contain any Lipschitz function composed as above (for example, Hadamard semi-differentiable functions [37], Whitney-stratifiable functions [9, 16], semi-algebraic functions etc.).*

The formal proof of the theorem appears in Appendix A in the supplementary material, but can be informally described as follows: First, we construct a Lipschitz function on $\mathbb{R}^d$, specified by a small vector $\mathbf{w}$, which resembles the norm function $\mathbf{x} \mapsto \|\mathbf{x}\|$ in "most" of $\mathbb{R}^d$, but with a "channel" leading away from a neighborhood of the origin in the direction of $\mathbf{w}$, and reaching a completely flat region (see Fig. 1). We emphasize that the graphical illustration is a bit misleading due to the low dimension: In high dimensions, the "channel" and flat region contain a vanishingly small portion of $\mathbb{R}^d$. This function has the property of having $\epsilon$-stationary points only in the flat region in the direction of $\mathbf{w}$, even though the function appears in most places like the norm function $\mathbf{x} \mapsto \|\mathbf{x}\|$. As a result, any oracle-based algorithm that doesn't happen to hit the vanishingly small region where the function differs from $\mathbf{x} \mapsto \|\mathbf{x}\|$, receives no information about $\mathbf{w}$, thus cannot determine where the $\epsilon$-stationary points lie. As a result, such an algorithm cannot return near-approximately-stationary points.

Unfortunately, the construction described so far does not work as-is, since the algorithm can always query sufficiently close to the origin (closer than roughly $\|\mathbf{w}\|$), where the gradients do provide information about $\mathbf{w}$. To prevent this, we compose the function with an algorithm-dependent affine mapping, which doesn't significantly change the function's properties, but ensures that the algorithm can never get too close to the (mapped) origin. We show that such an affine mapping must exist,

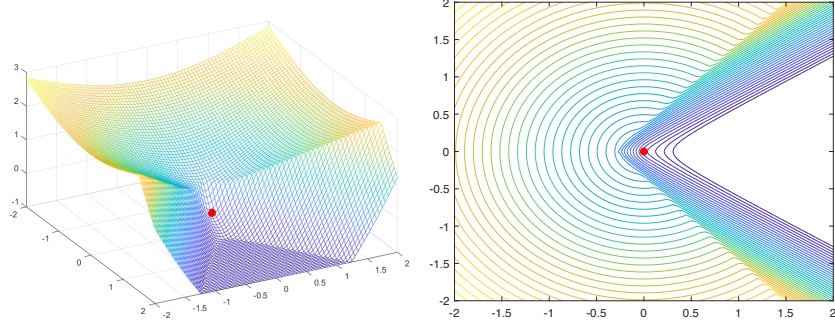

Figure 1: Mesh and Contour plot of the function $\mathbf{x} \mapsto \max\{-1, g_{\mathbf{w}}(\mathbf{x})\}$ on $\mathbb{R}^2$, where $\mathbf{w} = (0.3, 0)$ and $g_{\mathbf{w}}(\cdot)$ is defined in Lemma 2 (as part of the proof of Thm. 1). The origin is marked with a red dot. Best viewed in color.

based on standard oracle complexity results, which imply that oracle-based algorithms as above cannot get too close to the minimum of a generic convex quadratic function with a bounded number of queries. Using this mapping, and the useful property that $\mathbf{w}$ can be chosen arbitrarily small, leads to our theorem. The full proof details appear in Appendix A in the supplementary material.

**Remark 2** (Extension to higher-order algorithms). *Our proof approach is quite flexible, in the sense that for any algorithm, we really only need some function which cannot be optimized to arbitrarily high accuracy, composed with a "channel" construction as above. Since functions of this type also exist for algorithms employing higher-order derivatives beyond gradients (e.g., [2]), it is unlikely that such higher-order algorithms will circumvent our impossibility result.*

## 4 Smoothing Nonsmooth Nonconvex Functions

In this section, we turn to our second main contribution, examining the possibility of reducing nonsmooth nonconvex optimization to smooth nonconvex optimization, by running a smooth optimization method on a smooth approximation of the objective function. In what follows, we focus our discussion on 1-Lipschitz functions: This is without loss of generality, since if our objective is $L$-Lipschitz, we can simply rescale it by $L$ (and a Lipschitz assumption is always necessary if we wish to obtain a Lipschitz-gradient smooth approximation). Also, we focus on smoothing functions over all of $\mathbb{R}^d$ for simplicity, but our results and proofs easily extend to the case where we are only interested in smoothing over some bounded domain on which the function is Lipschitz.

For a nonsmooth *convex* function $f(\cdot)$, a well-known smoothing approach is proximal smoothing (also known as the Moreau envelope or Moreau-Yosida regularization [5]) defined as $P_\delta(f)$ where

$$P_\delta(f)(\mathbf{x}) := \min_{\mathbf{y}} \left( f(\mathbf{y}) + \frac{1}{2\delta}\|\mathbf{y} - \mathbf{x}\|^2 \right) . \tag{1}$$

By picking $\delta$ appropriately, $P_\delta(f)$ is an arbitrarily good smooth approximation of $f$: More formally, if $f$ is 1-Lipschitz, then for any $\epsilon > 0$, there exists a choice of $\delta = \Theta(\epsilon)$ such that $\|P_\delta(f) - f\|_\infty \leq \epsilon$, with the gradients of $P_\delta(f)$ being $\frac{1}{\epsilon}$-Lipschitz. This is essentially optimal, as no $\epsilon$-approximation can attain a gradient Lipschitz parameter better than $\Omega(1/\epsilon)$ (see Lemma 11 in Appendix E for a formal proof). Finally, computing gradients of $P_\delta(f)$ (which can then be fed into a gradient-based smooth optimization method) is feasible, given a solution to Eq. (1), which is a convex optimization problem and hence efficiently solvable in general.

Unfortunately, for nonconvex functions, proximal smoothing (or other smoothing methods from convex optimization) generally fails in producing smooth approximations. However, it turns out that similar guarantees can be obtained with a slightly more complicated procedure, known as *Lasry-Lions* regularization in the functional analysis literature [25, 3], which is essentially a double application of proximal smoothing combined with function flipping. One way to define it is as follows:

$$P_{\delta,\nu}(f)(\mathbf{x}) := -P_\delta(-P_\nu(f))(\mathbf{x}) = \max_{\mathbf{y}} \min_{\mathbf{z}} \left( f(\mathbf{z}) + \frac{1}{2\nu}\|\mathbf{z} - \mathbf{y}\|^2 - \frac{1}{2\delta}\|\mathbf{y} - \mathbf{x}\|^2 \right) .$$

Once more, if $f$ is 1-Lipschitz, then choosing $\delta, \nu = \Theta(\epsilon)$ appropriately, we get an $\epsilon$-accurate approximation of $f$, with gradients which are $\frac{c}{\epsilon}$-Lipschitz for some absolute constant $c$. However, unlike the convex case, implementing this smoothing involves solving a non-convex optimization problem, which may not be computationally tractable.

Alternatively, a very simple smoothing approach, which works equally well on convex and non-convex problems, is randomized smoothing, or equivalently, convolving the objective function with a smoothness-inducing density function. Formally, given the objective function $f$ and a distribution $P$, we define $\tilde{f}(\mathbf{x}) := \mathbb{E}_{\mathbf{y} \sim P}[f(\mathbf{x} + \mathbf{y})]$. In particular, letting $P$ be a uniform distribution on an origin-centered ball of radius $\epsilon$, the resulting function is an $\epsilon$-approximation of $f(\cdot)$, and its gradient Lipschitz parameter is $\frac{c\sqrt{d}}{\epsilon}$, where $c$ is an absolute constant and $d$ is the input dimension [19]. Moreover, given access to values and gradients of $f(\cdot)$, computing unbiased stochastic estimates of the values or gradients of $\tilde{f}(\cdot)$ is computationally very easy: We just sample a single $\mathbf{y} \sim P$, and return[3] $f(\mathbf{x} + \mathbf{y})$ or $\nabla f(\mathbf{x} + \mathbf{y})$. These stochastic estimates can then be plugged into stochastic methods for smooth optimization (see [19, 20]).

Comparing these two approaches, we see an interesting potential trade-off between the smoothness obtained and computational efficiency, summarized in the following table:

| | $\nabla \tilde{f}$ Lipschitz param. | Computationally Efficient? |
|---|---|---|
| Randomized Smoothing | $c \cdot \sqrt{d}/\epsilon$ | ✓ |
| Lasry-Lions Regularization | $c/\epsilon$ | ✗ |

In words, randomized smoothing is computationally efficient (unlike Lasry-Lyons regularization), but at the cost of a much larger gradient Lipschitz parameter. Since the iteration complexity of smooth optimization methods strongly depend on this Lipschitz parameter, it follows that in high-dimensional problems, we pay a high price for computational tractability in reducing nonsmooth to smooth problems. We emphasize that this is a real phenomenon, and not just an artifact of iteration complexity analysis, as we demonstrate in Appendix C.

This discussion leads to a natural question: Is this trade-off necessary, or perhaps there exist computationally efficient methods which can improve on randomized smoothing, in terms of the gradient Lipschitz parameter? Using an oracle complexity framework, we prove that this trade-off is indeed necessary (under mild assumptions), and that randomized smoothing is essentially an optimal method under the constraint of black-box access to the objective $f(\cdot)$, and a reasonable oracle complexity. We note that Duchi et al. [19] proved that the Lipschitz constant cannot be improved by simple randomized smoothing schemes, but here we consider a much larger class of possible methods.

### 4.1 Smoothing Algorithms

Before presenting our main result for this section, we need to carefully formalize what we mean by an efficient smoothing method, since "returning" a smooth approximating function over $\mathbb{R}^d$ is not algorithmically well-defined. Recalling the motivation to our problem, we want a method that given a nonsmooth objective function $f(\cdot)$, allows us to estimate values and gradients of a smooth approximation $\tilde{f}(\cdot)$ at arbitrary points, which can then be fed into standard black-box methods for smooth optimization (hence, we need a uniform approximation property). Moreover, for black-box optimization, it is desirable that this smoothing method operates in an oracle complexity framework, where it only requires local information about $f(\cdot)$ at various points. Finally, we are interested in controlling various parameters of the smoothing procedure, such as the degree of approximation, the smoothness of the resulting function, and the complexity of the procedure. In light of these considerations, a natural way to formalize smoothing methods is the following:

**Definition 1.** *An algorithm $\mathcal{A}$ is an $(L, \epsilon, T, M, r)$-smoother if for any 1-Lipschitz function $f$ on $\mathbb{R}^d$, there exists a differentiable function $\tilde{f}$ on $\mathbb{R}^d$ with the following properties:*

*1. $\|f - \tilde{f}\|_\infty \leq \epsilon$, and $\nabla \tilde{f}$ is $L$-Lipschitz.*

*2. Given any $\mathbf{x} \in \mathbb{R}^d$, the algorithm produces a (possibly randomized) query sequence $\mathbf{x}_1, \ldots, \mathbf{x}_T \in \{\mathbf{y} : \|\mathbf{y} - \mathbf{x}\| \leq r\}$, of the form $\mathbf{x}_{i+1} = \mathcal{A}^{(i)}\left(\xi, \mathbf{x}, \mathsf{O}_f^\infty(\mathbf{x}_1), \ldots, \mathsf{O}_f^\infty(\mathbf{x}_i)\right)$, where $\mathcal{A}^{(i)}$ maps*

---

[3]By Rademacher's theorem, $f$ is differentiable almost everywhere hence $\nabla f(\mathbf{x} + \mathbf{y})$ exists almost surely.

*all the previous information gathered by the queries to a new query, possibly based on a draw of some random variable[4] $\xi$. Finally, the algorithm produces a vector*

$$\mathcal{A}(f, \mathbf{x}) := \mathcal{A}^{(out)}\left(\xi, \mathbf{x}, \mathsf{O}_f^\infty(\mathbf{x}_1), \ldots, \mathsf{O}_f^\infty(\mathbf{x}_T)\right) ,$$

*where $\mathcal{A}^{(out)}$ is some mapping to $\mathbb{R}^d$, such that*

$$\left\|\mathbb{E}_\xi\left[\mathcal{A}(f, \mathbf{x})\right] - \nabla\tilde{f}(\mathbf{x})\right\| \le \epsilon \quad \text{and} \quad \Pr_\xi\left[\|\mathcal{A}(f, \mathbf{x})\| \le M\right] = 1 . \tag{2}$$

Some comments about this definition are in order. First, the definition is only with respect to the ability of the algorithm to approximate gradients of $\tilde{f}(\cdot)$: It is quite possible that the algorithm also has additional output (such as an approximation of the value of $\tilde{f}(\cdot)$), but this is not required for our results. Second, we do not require the algorithm to return $\nabla\tilde{f}(\mathbf{x})$: It is enough that the expectation of the vector output is close to it (up to $\epsilon$). This formulation captures both deterministic optimization-type methods (such as Lasry-Lyons regularization, where in general we can only hope to solve the auxiliary optimization problem up to some finite precision) as well as stochastic methods (such as randomized smoothing, which returns $\nabla\tilde{f}(\mathbf{x})$ in expectation). Third, we assume that the queries returned by the algorithm lie at a bounded distance $r$ from the input point $\mathbf{x}$. In the context of randomized smoothing, this corresponds (for example) to using a uniform distribution over a ball of radius $r$ centered on $\mathbf{x}$. As we discuss later on, some assumption on the magnitude of the queries is necessary for our proof technique. However, requiring almost-sure boundedness is merely for simplicity, and it can be replaced by a high-probability bound with appropriate tail assumptions (e.g., if we are performing randomized smoothing with a Gaussian distribution), at the cost of making the analysis a bit more complicated.

**Remark 3.** *We are mostly interested in the following parameter regimes:*

- *$T = poly\left(d, L, \epsilon^{-1}\right)$, essentially meaning that a single call to $\mathcal{A}$ is computable in a reasonable amount of time.*

- *$M = poly(L)$. As we formally prove in Lemma 12 in Appendix E, if we require $\tilde{f}$ to approximate $f$ and also have L-Lipschitz gradients, we must have $\|\nabla\tilde{f}(x)\| = \mathcal{O}(L)$. In particular, whenever $M$ is sufficiently larger than $L$, Eq. (2) is interchangeable with the seemingly more natural condition $\Pr_\xi\left[\|\mathcal{A}(f, \mathbf{x}) - \nabla\tilde{f}(\mathbf{x})\| \le M\right] = 1.$*

- *$r = \mathcal{O}(\epsilon)$. If we are interested in smoothing a 1-Lipschitz, nonconvex function up to $\epsilon$ accuracy around a given point $\mathbf{x}$, we generally expect that only its $\mathcal{O}(\epsilon)$-neighborhood will convey useful information for the smoothing process. We note that this is indeed satisfied by randomized smoothing (with a uniform distribution around a radius-$\epsilon$ ball, or with high probability if we use a Gaussian distribution), as well as Lasry-Lyons regularization (in the sense that the smooth approximation at $\mathbf{x}$ does not change if we alter the function arbitrarily outside an $\mathcal{O}(\epsilon)$-neighborhood of $\mathbf{x}$).*

*We consider all three of the above to be quite permissive. In particular, notice that randomized smoothing over a ball satisfies the much stronger $T = 1$, $M = 1$, $r = \epsilon$.*

Our result will require the assumption that the smoothing algorithm $\mathcal{A}$ is *translation invariant with respect to constant functions*, in the sense that it treats all constant functions and regions of the input space in the same manner. In order to formalize this property, first notice that if $f$ is a constant function then the all-derivatives oracle $\mathsf{O}_f^\infty(\mathbf{x})$ does not depend on $\mathbf{x}$. Hence, for a constant function $f$ we can use the abbreviation $\mathsf{O}_f^\infty$ without specifying the specific input point. We use this to formalize our desired translation-invariance property as follows:

**Definition 2.** *A smoothing algorithm $\mathcal{A}$ satisfies* TICF *(translation invariance w.r.t. constant functions) if for any two constant functions $f, g$, any $\mathbf{x} \in \mathbb{R}^d$ and $i \in [T]$, and any realization of $\xi$,*

$$\mathcal{A}^{(i)}\left(\xi, \mathbf{x}, \mathsf{O}_f^\infty, \ldots, \mathsf{O}_f^\infty\right) = \mathcal{A}^{(i)}\left(\xi, \mathbf{0}, \mathsf{O}_g^\infty, \ldots, \mathsf{O}_g^\infty\right) + \mathbf{x} , \tag{3}$$

*and*

$$\mathcal{A}^{(out)}\left(\xi, \mathbf{x}, \mathsf{O}_f^\infty, \ldots, \mathsf{O}_f^\infty\right) = \mathcal{A}^{(out)}\left(\xi, \mathbf{0}, \mathsf{O}_g^\infty, \ldots, \mathsf{O}_g^\infty\right) .$$

---

[4]We assume nothing about $\xi$, allowing the algorithm to utilize an arbitrary amount of randomness.

In other words, if instead of a constant function $f$ and an input point $\mathbf{x}$, we pick some other constant function $g$ and the origin, the distribution of the algorithm's sequence of queries remain the same (up to a shift by $-\mathbf{x}$), and the gradient estimate returned by the algorithm remains the same. We consider this to be a mild and natural assumption, which is clearly satisfied by standard smoothing techniques.

## 4.2 Main result

With these definitions in hand, we are finally ready to present our main result for this section:

**Theorem 2.** *Let $\mathcal{A}$ be an $(L, \epsilon, T, M, r)$-smoother which satisfies TICF. Then there exist absolute constants $c_1, c_2 > 0$ such that*

$$L\sqrt{\log\left((M+1)(T+1)\right)} \geq c_1 \cdot \frac{\sqrt{d}}{r}\left(c_2 - \epsilon\right) \ . \tag{4}$$

This theorem holds for general values of the parameters $L, \epsilon, T, M, r$. Concretely, for parameter regimes of interest (see Remark 3) we have the following corollary:

**Corollary 1.** *Suppose that the accuracy parameter satisfies $\epsilon \leq c_2/2$. Then any smoothing algorithm which makes at most $T = poly(d, L, \epsilon^{-1})$ queries at a distance at most $r = \mathcal{O}(\epsilon)$ from the input point, and returns vectors of norm at most $M = poly(d, L, \epsilon^{-1})$, must correspond to a smooth approximation $\tilde{f}(\cdot)$ with Lipschitz gradient parameter at least $L = \tilde{\Omega}(\sqrt{d}/\epsilon)$.*

We note that the lower bound on $L$ in this corollary matches (up to logarithmic factors) the upper bound attained by randomized smoothing. This implies that at least under our framework and assumptions, randomized smoothing is an essentially optimal efficient smoothing method.

Another implication of Thm. 2 is that even if we relax our assumption that $r = \mathcal{O}(\epsilon)$, then as long as $r$ does not scale with the dimension $d$, the gradient Lipschitz parameter of any efficient smoothing algorithm must scale with the dimension (even though there exist dimension-free smooth approximations, as evidenced by Lasry-Lyons regularization):

**Corollary 2.** *Fix any accuracy parameter $\epsilon \leq c_2/2$, and any $r > 0$. Then as long as the number of queries is $T = poly(d)$ and the output is of size $M = poly(d)$, we must have $L \geq \tilde{\Omega}(\sqrt{d})$.*

A third corollary of our theorem is that (perhaps unsurprisingly), there is no way to implement Lasry-Lions regularization efficiently in an oracle complexity framework:

**Corollary 3.** *If $\epsilon \leq c_2/2$ and $M = poly(d)$, then any smoothing algorithm for which $\tilde{f}(\cdot)$ corresponds to the Lasry-Lions regularization (which satisfies $L = \mathcal{O}(1)$) must use a number of queries $T = \exp(\tilde{\Omega}(d))$.*

**Remark 4** (Dependence on $M$). *Our definition of a smoothing algorithm focuses on the expectation of the algorithm's output. This leads to a logarithmic dependence on $M$ (an upper bound on the algorithm's output) in Thm. 2, since in the proof we need to bound the influence of exponentially-small-probability events on the expectation. It is plausible that the dependence on $M$ can be eliminated altogether, by changing the definition of a smoothing algorithm to focus on the expectation of its output, conditioned on all but highly-improbably events. However, that would complicate the definition and our proof.*

We will now outline the proof idea, which is formally presented in Appendix A. Consider a one dimensional monotonically increasing function $g$, which is locally constant at a $\Omega(1/\sqrt{d})$ neighborhood of a grid $\Delta = \{0, \delta_1, \ldots, \delta_K\} \subset [0, 1]$, with $K$ roughly of order $\sqrt{d}$ (see Fig. 2 in Appendix A). We define $f(\mathbf{x}) = g(\mathbf{w}^\top \mathbf{x})$, where $\mathbf{w} \in \mathcal{S}^{d-1}$ is a uniformly random unit vector. We note that $f$ is a simple function, easily implemented by (say) a one-hidden layer ReLU neural network. We proceed to analyze what happens when a smoothing algorithm is given points of the form $\delta_i \mathbf{w}$, $\delta_i \in \Delta$. Since $\mathbf{w}$ is random, and the algorithm is assumed to be translation invariant, it can be shown (via a concentration of measure argument) that the algorithm is overwhelmingly likely to produce queries in directions which all have $\tilde{\mathcal{O}}(1/\sqrt{d})$ correlation with $\mathbf{w}$, as long as the number of queries is polynomial. Consequently, with high probability, the queries all lie in a region where the function $f(\cdot)$ is flat, and the algorithm cannot distinguish between it and a constant function. By the translation-invariance property, this implies that the gradient estimates $\nabla \tilde{f}(\delta_i \mathbf{w})$ must be of small norm, uniformly for all

$\delta_i \mathbf{w}$. Combining the observation that $\nabla \tilde{f}(\cdot)$ is small along order-of-$\sqrt{d}$-many points between $\mathbf{0}$ and the unit vector $\mathbf{w}$, together with the fact that $\nabla \tilde{f}(\cdot)$ is $L$-Lipschitz, we can derive an upper bound on how much $\tilde{f}(\cdot)$ can increase along the line segment between $\mathbf{0}$ and $\mathbf{w}$, roughly on the order of $L/\sqrt{d}$. On the other hand, $\tilde{f}(\cdot)$ is an approximation of $f$, which has a constant increase between $\mathbf{0}$ and $\mathbf{w}$. Overall this allows us to deduce a lower bound on $L$ scaling as $\sqrt{d}$, which results in the theorem.

## 5   Discussion

In this paper, we studied the problem of nonconvex, nonsmooth optimization from an oracle complexity perspective, and provided two main results: One (in Sec. 3) is an impossibility result for efficiently getting near approximately-stationary points, and the second (in Sec. 4) proving an inherent trade-off between oracle complexity and the smoothness parameter when smoothing nonsmooth functions. The second result also establishes the optimality of randomized smoothing as an efficient smoothing method, under mild assumptions.

Our work leaves open several questions. First, at a more technical level, there is the question of whether some or all of our assumptions can be relaxed. In particular, the result in Sec. 3 currently holds for deterministic and zero-respecting algorithms. Although this is already a very broad class, one may ask whether it can be made to hold for all algorithms. We believe this is true, and (as we remark in Remark 5) will directly follow from a more general form of first-order oracle complexity bounds for quadratics. As for the smoothness result in Sec. 4, it currently requires the algorithm to be translation invariant w.r.t. constant functions, as well as querying at some bounded distance from the input point $\mathbf{x}$. We conjecture that the translation invariance assumption can be relaxed, possibly by a suitable reduction that shows that any smoothing algorithm can be converted to a translation invariant one. However, how to formally perform this remains unclear at the moment. As to the bounded distance of the queries, it is currently an essential assumption for our proof technique, which relies on a function which looks "locally" constant at many different points, but is globally non-constant, and this can generally be determined by querying far enough away from the input point (even along some random direction). Thus, relaxing this assumption may necessitate a different proof technique.

Another open question is whether randomized smoothing can be "derandomized": Our results indicate that the gradient Lipschitz parameter of the smooth approximation cannot be improved, but leave open the possibility of an efficient method returning the actual gradients of some smooth approximation (up to machine precision), in contrast to randomized smoothing which only provides noisy stochastic estimates of the gradient. These can then be plugged into smooth optimization methods which assume access to the exact gradients (rather than noisy stochastic estimates), generally improving the resulting iteration complexity. We note that naively computing the exact gradient of $\tilde{f}(\cdot)$ arising from randomized smoothing is infeasible in general, as it involves a high-dimensional integral.

At a more general level, our work leaves open the question of what is the "right" metric to study for nonsmooth-nonconvex optimization, where neither minimizing optimization error nor finding approximately-stationary points is feasible. In this paper, we show that the goal of getting near approximately stationary points is not feasible, at least in the worst case, whereas smoothing can be done efficiently, but not in a dimension-free manner. Can we find other compelling goals to consider? One very appealing notion is the $(\delta, \epsilon)$-stationarity of [37] that we mentioned in the introduction, which comes with clean, finite-time and dimension-free guarantees. Our negative result in Thm. 1 provides further motivation to consider it, by showing that a natural variation of this notion will not work. However, as we discuss in Appendix B in the supplementary material, we need to accept that this stationarity notion can have unexpected behavior, and there exist cases where it will not resemble a stationary point in any intuitive sense. In any case, using an oracle complexity framework to study this and other potential metrics for nonsmooth nonconvex optimization, which combine computational efficiency and finite-time guarantees, remains an interesting direction for future research.

**Broader impact.** In this work we discuss several approaches for providing formal guarantees when optimizing nonsmooth nonconvex functions, and shed light on their inherent limitations and trade-offs in an oracle complexity framework. Such optimization objectives arise when training modern deep neural networks, a practice which is currently subject to many heuristics. We believe a better theoretical understanding of the field will eventually affect practice and help develop sound deep learning based technology. We find it difficult to provide more specific potential negative societal impact, other than the well known potential negative implications of such technology as a whole.

**Funding disclosure.** This research is supported by the European Research Council (ERC) grant 754705.

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
