## A  Proofs

### A.1  Proof of Thm. 1

We begin by stating the following theorem, which follows from well-known results in oracle complexity (see [31, 27]):

**Theorem 3.** *For any $T > 1$, any algorithm in $\mathcal{A}_{det} \cup \mathcal{A}_{zr}$ and any dimension $d \geq 2T$, there is a vector $\mathbf{x}^* \in \mathbb{R}^d$ (where $\|\mathbf{x}^*\| \leq \frac{1}{2}$) and a positive definite matrix $M \in \mathbb{R}^{d \times d}$ (with minimal and maximal eigenvalues satisfying $\frac{1}{2} \leq \lambda_{\min}(M) \leq \lambda_{\max}(M) \leq 1$), such that the iterates $\mathbf{x}_1, \ldots, \mathbf{x}_T$ produced by the algorithm when ran on the strictly convex quadratic function $f(\mathbf{x}) := (\mathbf{x} - \mathbf{x}^*)^\top M(\mathbf{x} - \mathbf{x}^*)$ satisfy*

$$\min_{t \in \{1, \ldots, T\}} \|\mathbf{x}_t - \mathbf{x}^*\| \geq \exp(-T).$$

For completeness, we provide a self-contained proof in Appendix D. Basically, the theorem states that for any algorithm in $\mathcal{A}_{det} \cup \mathcal{A}_{zr}$, there is a well-conditioned[5] but still "relatively hard" strictly convex quadratic function, whose minimum cannot be detected with accuracy better than $\exp(-T)$.

**Remark 5** (Extension to any algorithm). *Up to the constants, a lower bound as in Thm. 3 is widely considered to hold (with high-probability) for* all *algorithms based on a first-order oracle, not just for deterministic or zero-respecting ones (see [27, 33]). In that case, our Thm. 1 can be easily extended to apply to all oracle-based algorithms which utilize function values and gradients, since the only point in the proof where we really need to restrict the algorithm class is in Thm. 3. Unfortunately, we are not aware of a result in the literature which quite states this, explicitly and in the required form. For example, there are algorithm-independent lower bounds which rely on non-quadratic functions [35], or apply to quadratics, but not in a regime where $\lambda_{\max}(M)/\lambda_{\min}(M)$ is a constant as in our case [33].*

Given the theorem, our first step will be to reduce it to a hardness result for optimizing convex Lipschitz functions of the form $\mathbf{x} \mapsto \|M^{1/2}(\mathbf{x} - \mathbf{x}^*)\|$:

**Lemma 1.** *For any algorithm in $\mathcal{A}_{det} \cup \mathcal{A}_{zr}$, any $T > 1$ and any dimension $d \geq 2T$, there is a vector $\mathbf{x}^* \in \mathbb{R}^d$ (where $\|\mathbf{x}^*\| \leq \frac{1}{2}$) and a positive definite matrix $M \in \mathbb{R}^{d \times d}$ (with $\frac{1}{2} \leq \lambda_{\min}(M) \leq \lambda_{\max}(M) \leq 1$), such that the convex function*

$$\hat{f}(\mathbf{x}) := \|M^{1/2}(\mathbf{x} - \mathbf{x}^*)\|$$

*satisfies the following:*

- *$\hat{f}(\cdot)$ is $\frac{1}{\sqrt{2}}$-Lipschitz, and $\hat{f}(\mathbf{0}) \leq \frac{1}{2}$.*

- *If we run the algorithm on $\hat{f}(\cdot)$, then $\min_{t \in \{1, \ldots, T\}} \|\mathbf{x}_t - \mathbf{x}^*\| \geq \exp(-T)$.*

*Proof.* We will start with the second bullet. Fix some algorithm $A$ in $\mathcal{A}_{det} \cup \mathcal{A}_{zr}$, and assume by contradiction that for any $\mathbf{x}^*, M$ satisfying the conditions in the lemma, the algorithm runs on $\hat{f}(\cdot)$ and produces iterates such that $\min_{t \in \{1, \ldots, T\}} \|\mathbf{x}_t - \mathbf{x}^*\| < \exp(-T)$ (either deterministically if the algorithm is deterministic, or for some realization of its random coin flips, if it is randomized). But then, we argue that given access to gradients and values of $f(\mathbf{x}) := \hat{f}^2(\mathbf{x}) = (\mathbf{x} - \mathbf{x}^*)^\top M(\mathbf{x} - \mathbf{x}^*)$, we can use $A$ to specify *another* algorithm in $\mathcal{A}_{det} \cup \mathcal{A}_{zr}$ that runs on $f(\cdot)$ and produces points $\mathbf{x}_1, \ldots, \mathbf{x}_T$ such that $\min_{t \in \{1, \ldots, T\}} \|\mathbf{x}_t - \mathbf{x}^*\| < \exp(-T)$, contradicting Thm. 3. To see why, note that given access to an oracle returning values and gradients of $f(\cdot)$ at $\mathbf{x}$, we can simulate an oracle returning gradients and values of $\hat{f}(\cdot)$ at $\mathbf{x}$ via the easily-verified formulaes

$$\hat{f}(\mathbf{x}) = \sqrt{f(\mathbf{x})} \quad \text{and} \quad \nabla\hat{f}(\mathbf{x}) = \frac{1}{2\sqrt{f(\mathbf{x})}}\nabla f(\mathbf{x})$$

(and for $\mathbf{x} = \mathbf{x}^*$ where $\hat{f}(\cdot)$ is not differentiable, we can just return the value 0 and the generalized gradient set at $\mathbf{0}$). We then feed the responses of this simulated oracle to $A$, and get the resulting

---

[5]In the sense that $\lambda_{\max}(M)/\lambda_{\min}(M) \leq 2$.

$\mathbf{x}_1, \ldots, \mathbf{x}_T$. This give us a new algorithm $A'$, which is easily verified to be in $\mathcal{A}_{det} \cup \mathcal{A}_{zr}$ if the original algorithm $A$ is in $\mathcal{A}_{det} \cup \mathcal{A}_{zr}$.

It remains to prove the second bullet in the lemma. First, we have $\hat{f}(\mathbf{0}) = \|M^{1/2}\mathbf{x}^*\| \leq \sqrt{\|M\|}\|\mathbf{x}^*\| \leq \frac{1}{2}$. Second, we note that for any $\mathbf{x} \neq \mathbf{x}^*$, $\hat{f}(\cdot)$ is differentiable and satisfies

$$\|\nabla\hat{f}(\mathbf{x})\| = \frac{\|M(\mathbf{x}-\mathbf{x}^*)\|}{2\|M^{1/2}(\mathbf{x}-\mathbf{x}^*)\|} \leq \frac{\lambda_{\max}(M)\cdot\|\mathbf{x}-\mathbf{x}^*\|}{2\sqrt{\lambda_{\min}(M)}\cdot\|\mathbf{x}-\mathbf{x}^*\|} = \frac{\lambda_{\max}(M)}{2\sqrt{\lambda_{\min}(M)}} ,$$

which by the conditions on $M$, is at most $\frac{1}{2\sqrt{1/2}} = \frac{1}{\sqrt{2}}$. $\qquad\square$

Next, we define a function $g(\cdot)$ with two properties: It is identical to $\mathbf{x} \mapsto \|\mathbf{x}\|$ in parts of $\mathbb{R}^d$ (in fact, as we will see later, in "almost" all of $\mathbb{R}^d$), yet unlike the function $\mathbf{x} \mapsto \|\mathbf{x}\|$, it has no stationary points, or even $\epsilon$-stationary points.

**Lemma 2.** *Fix some vector $\mathbf{w} \neq \mathbf{0}$ in $\mathbb{R}^d$, and define the function*

$$g_{\mathbf{w}}(\mathbf{x}) := \|\mathbf{x}\| - \left[4\bar{\mathbf{w}}^\top(\mathbf{x}+\mathbf{w}) - 2\|\mathbf{x}+\mathbf{w}\|\right]_+ ,$$

*where $\bar{\mathbf{u}} := \mathbf{u}/\|\mathbf{u}\|$ for any vector $\mathbf{u}$, and $[v]_+ := \max\{v,0\}$. Then $g_{\mathbf{w}}(\cdot)$ is 7-Lipschitz, and has no $\epsilon$-stationary points for any $\epsilon < \frac{1}{\sqrt{2}}$ .*

*Proof.* In the proof, we will drop the $\mathbf{w}$ subscript and refer to $g_{\mathbf{w}}(\cdot)$ as $g(\cdot)$.

The functions $\mathbf{x} \mapsto \|\mathbf{x}\|$, $\mathbf{x} \mapsto 4\bar{\mathbf{w}}^\top(\mathbf{x}+\mathbf{w})$, $\mathbf{x} \mapsto 2\|\mathbf{x}+\mathbf{w}\|$ and $x \mapsto \max\{0,x\}$ are respectively 1-Lipschitz, 4-Lipschitz, 2-Lipschitz and 1-Lipschitz, from which it immediately follows that $g(\cdot)$ is $1+4+2 = 7$ Lipschitz. Thus, it only remains to show that $g(\cdot)$ has no $\epsilon$-stationary points.

It is easily seen that the function $g(\cdot)$ is not differentiable at only 3 possible regions: (1) $\mathbf{x} = \mathbf{0}$, (2) $\mathbf{x} = -\mathbf{w}$, and (3) $\{\mathbf{x} : 4\bar{\mathbf{w}}^\top(\mathbf{x}+\mathbf{w}) - 2\|\mathbf{x}+\mathbf{w}\| = 0\}$ (or equivalently, $\{\mathbf{x} : \bar{\mathbf{w}}^\top(\overline{\mathbf{x}+\mathbf{w}}) = \frac{1}{2}\}$ if we exclude $\mathbf{x} = -\mathbf{w}$), which are all measure-zero sets in $\mathbb{R}^d$. At any other point, $g(\cdot)$ is differentiable and the gradient satisfies

$$\nabla g(\mathbf{x}) = \bar{\mathbf{x}} - \mathbf{1}_{\bar{\mathbf{w}}^\top(\overline{\mathbf{x}+\mathbf{w}})>\frac{1}{2}} \cdot (4\bar{\mathbf{w}} - 2(\overline{\mathbf{x}+\mathbf{w}})) .$$

Moreover, at those differentiable points, if $\bar{\mathbf{w}}^\top(\overline{\mathbf{x}+\mathbf{w}}) < \frac{1}{2}$ then

$$\|\nabla g(\mathbf{x})\| = \|\bar{\mathbf{x}}\| = 1 ,$$

and if $\bar{\mathbf{w}}^\top(\overline{\mathbf{x}+\mathbf{w}}) > \frac{1}{2}$, then by the triangle inequality,

$$\|\nabla g(\mathbf{x})\| = \|\bar{\mathbf{x}} - (4\bar{\mathbf{w}} - 2(\overline{\mathbf{x}+\mathbf{w}}))\| = \|4\bar{\mathbf{w}} - 2(\overline{\mathbf{x}+\mathbf{w}}) - \bar{\mathbf{x}}\|$$
$$\geq 4\|\bar{\mathbf{w}}\| - 2\|\overline{\mathbf{x}+\mathbf{w}}\| - \|\bar{\mathbf{x}}\| = 4 - 2 - 1 = 1 .$$

Thus, no differentiable point of $g$ is even 0.99-stationary. It remains to show that even the non-differentiable points of $g$ are not $\epsilon$-stationary for any $\epsilon < \frac{1}{\sqrt{2}}$. To do so, we will use the facts that $\partial(g_1 + g_2) \subseteq \partial g_1 + \partial g_2$, and that if $g_1$ is univariate, $\partial(g_1 \circ g_2)(\mathbf{x}) \subseteq \mathrm{conv}\{r_1\mathbf{r}_2 : r_1 \in \partial g_1(g_2(\mathbf{x})), \mathbf{r}_2 \in \partial g_2(\mathbf{x})\}$ (see [14]).

- At $\mathbf{x} = \mathbf{0}$, we have

$$\partial g(\mathbf{x}) \subseteq \mathrm{conv}\{\mathbf{u} - 2\bar{\mathbf{w}} : \|\mathbf{u}\| \leq 1\} = \{\mathbf{u} - 2\bar{\mathbf{w}} : \|\mathbf{u}\| \leq 1\} .$$

  Any element in this set has a norm of $\|\mathbf{u} - 2\bar{\mathbf{w}}\| = \|2\bar{\mathbf{w}} - \mathbf{u}\| \geq 2\|\bar{\mathbf{w}}\| - \|\mathbf{u}\| \geq 2 - 1 = 1$ by the triangle inequality. Thus, $\mathbf{x} = \mathbf{0}$ is not $\epsilon$-stationary for any $\epsilon < 1$.

- At $\mathbf{x} = -\mathbf{w}$, we have

$$\partial g(\mathbf{x}) \subseteq \mathrm{conv}\{-\bar{\mathbf{w}} - v \cdot (4\bar{\mathbf{w}} - 2\mathbf{u}) : v \in [0,1], \|\mathbf{u}\| \leq 1\}$$
$$= \mathrm{conv}\{2v\mathbf{u} - (1+4v)\bar{\mathbf{w}} : v \in [0,1], \|\mathbf{u}\| \leq 1\} .$$

  For any element in the set $\{2v\mathbf{u} - (1+4v)\bar{\mathbf{w}} : v \in [0,1], \|\mathbf{u}\| \leq 1\}$ (corresponding to some $v, \mathbf{u}$), its inner product with $-\bar{\mathbf{w}}$ is

$$-2v\bar{\mathbf{w}}^\top\mathbf{u} + (1+4v) \geq -2v + (1+4v) \geq 1 .$$

  Thus, any element in the convex hull of this set, which contains $\partial g(\mathbf{x})$, has an inner product of at least 1 with $-\bar{\mathbf{w}}$. Since $-\bar{\mathbf{w}}$ is a unit vector, it follows that the norm of any element in $\partial g(\mathbf{x})$ is at least 1, so this point is not $\epsilon$-stationary for any $\epsilon < 1$.

- At any $\mathbf{x}$ in the set $\{\mathbf{x} : \bar{\mathbf{w}}^\top(\overline{\mathbf{x} + \mathbf{w}}) = \frac{1}{2}\} \setminus \{\mathbf{0}, -\mathbf{w}\}$, we have

$$\partial g(\mathbf{x}) \subseteq \text{conv}\left\{\bar{\mathbf{x}} - v \cdot (4\bar{\mathbf{w}} - 2(\overline{\mathbf{x} + \mathbf{w}})) : v \in [0, 1]\right\}$$
$$= \left\{\bar{\mathbf{x}} - v \cdot (4\bar{\mathbf{w}} - 2(\overline{\mathbf{x} + \mathbf{w}})) : v \in [0, 1]\right\}$$
$$= \left\{\left(\frac{1}{\|\mathbf{x}\|} + \frac{2v}{\|\mathbf{x} + \mathbf{w}\|}\right)\mathbf{x} - \left(\frac{4v}{\|\mathbf{w}\|} - \frac{2v}{\|\mathbf{x} + \mathbf{w}\|}\right)\mathbf{w} : v \in [0, 1]\right\}. \quad (5)$$

Let $\mathbf{x} = \mathbf{x}_| + \mathbf{x}_\perp$, where $\mathbf{x}_\perp = (I - \bar{\mathbf{w}}\bar{\mathbf{w}}^\top)\mathbf{x}$ is the component of $\mathbf{x}$ orthogonal to $\mathbf{w}$, and $\mathbf{x}_| \in \text{span}(\mathbf{w})$. Thus, any element in $\partial g(\mathbf{x})$ can be written as

$$\left(\frac{1}{\|\mathbf{x}\|} + \frac{2v}{\|\mathbf{x} + \mathbf{w}\|}\right)\mathbf{x}_\perp + a \cdot \mathbf{w}$$

for some scalar $a$. Since $\mathbf{w}$ is orthogonal to $\mathbf{x}_\perp$, the norm of this element is at least

$$\left(\frac{1}{\|\mathbf{x}\|} + \frac{2v}{\|\mathbf{x} + \mathbf{w}\|}\right)\|\mathbf{x}_\perp\| \geq \frac{1}{\|\mathbf{x}\|} \cdot \|\mathbf{x}_\perp\|.$$

Noting that

$$\|\mathbf{x}_\perp\|^2 = \mathbf{x}^\top(I - \bar{\mathbf{w}}\bar{\mathbf{w}}^\top)^2\mathbf{x} = \mathbf{x}^\top(I - \bar{\mathbf{w}}\bar{\mathbf{w}}^\top)\mathbf{x} = \|\mathbf{x}\|^2 - (\bar{\mathbf{w}}^\top\mathbf{x})^2 = \|\mathbf{x}\|^2(1 - (\bar{\mathbf{w}}^\top\bar{\mathbf{x}})^2)$$

and plugging into the above, it follows that the norm is at least $\sqrt{(1 - (\bar{\mathbf{w}}^\top\bar{\mathbf{x}})^2)}$.

Now, let us suppose that there exists an element in $\partial g(\mathbf{x})$ with norm at most $\epsilon$. By the above, it follows that

$$\sqrt{(1 - (\bar{\mathbf{w}}^\top\bar{\mathbf{x}})^2)} \leq \epsilon. \quad (6)$$

However, we will show that for any $\epsilon < \frac{1}{\sqrt{2}}$, we must arrive at a contradiction, which implies that $\mathbf{x}$ cannot be $\epsilon$-stationary for $\epsilon < \frac{1}{\sqrt{2}}$. To that end, let us consider two cases:

- If $\bar{\mathbf{w}}^\top\bar{\mathbf{x}} \leq 0$, then by Eq. (6), we must have $\bar{\mathbf{w}}^\top\bar{\mathbf{x}} \leq -\sqrt{1 - \epsilon^2}$. But then, for any $\mathbf{u} \in \partial g(\mathbf{x})$, by Eq. (5) and our assumption that $\bar{\mathbf{w}}^\top(\overline{\mathbf{x} + \mathbf{w}}) = \frac{1}{2}$,

$$\bar{\mathbf{w}}^\top\mathbf{u} = \bar{\mathbf{w}}^\top\bar{\mathbf{x}} - v \cdot \left(4 - 2 \cdot \frac{1}{2}\right) \leq -\sqrt{1 - \epsilon^2} - 3v \leq -\sqrt{1 - \epsilon^2}.$$

  This implies that $\|\mathbf{u}\| \geq \sqrt{1 - \epsilon^2}$ for any $\mathbf{u} \in \partial g(\mathbf{x})$. Thus, if there was some $\mathbf{u} \in \partial g(\mathbf{x})$ with norm at most $\epsilon$, we get that $\epsilon \geq \sqrt{1 - \epsilon^2}$, which cannot hold if $\epsilon < \frac{1}{\sqrt{2}}$.

- If $\bar{\mathbf{w}}^\top\bar{\mathbf{x}} > 0$, then by Eq. (6), we have $\bar{\mathbf{w}}^\top\bar{\mathbf{x}} \geq \sqrt{1 - \epsilon^2}$. Hence,

$$\bar{\mathbf{w}}^\top(\mathbf{x} + \mathbf{w}) \geq \|\mathbf{x}\|\sqrt{1 - \epsilon^2} + \|\mathbf{w}\| \geq (\|\mathbf{x}\| + \|\mathbf{w}\|)\sqrt{1 - \epsilon^2} \geq \|\mathbf{x} + \mathbf{w}\|\sqrt{1 - \epsilon^2}.$$

  However, dividing both sides by $\|\mathbf{x} + \mathbf{w}\|$, we get that $\bar{\mathbf{w}}^\top(\overline{\mathbf{x} + \mathbf{w}}) \geq \sqrt{1 - \epsilon^2}$. If $\epsilon < \frac{1}{\sqrt{2}}$, it follows that $\bar{\mathbf{w}}^\top(\overline{\mathbf{x} + \mathbf{w}}) > \frac{1}{\sqrt{2}}$, which contradicts our assumption that $\mathbf{x}$ satisfies $\bar{\mathbf{w}}^\top(\overline{\mathbf{x} + \mathbf{w}}) = \frac{1}{2}$.

$\square$

**Lemma 3.** *Fix any algorithm in $\mathcal{A}_{det} \cup \mathcal{A}_{zr}$, any $T > 1$ and any $d \geq 2T$. Define the function*

$$h_{\mathbf{w}}(\mathbf{x}) := \max\{-1, g_{\mathbf{w}}(M^{1/2}(\mathbf{x} - \mathbf{x}^*))\}$$
$$= \max\left\{-1, \|M^{1/2}(\mathbf{x} - \mathbf{x}^*)\| - \left[4\bar{\mathbf{w}}^\top(M^{1/2}(\mathbf{x} - \mathbf{x}^*) + \mathbf{w}) - 2\|M^{1/2}(\mathbf{x} - \mathbf{x}^*) + \mathbf{w}\|\right]_+\right\},$$

*where $M, \mathbf{w}^*$ are as defined in Lemma 1, $g_{\mathbf{w}}(\cdot)$ is as defined in Lemma 2, and $\mathbf{w}$ is a vector of norm $\frac{1}{300}\exp(-T)$ in $\mathbb{R}^d$. Then:*

- *$h_{\mathbf{w}}(\cdot)$ is 7-Lipschitz, and satisfies $h_{\mathbf{w}}(\mathbf{0}) - \inf_{\mathbf{x}} h_{\mathbf{w}}(\mathbf{x}) \leq \frac{3}{2}$.*

- *Any $\epsilon$-stationary point $\mathbf{x}$ of $h_{\mathbf{w}}(\cdot)$ for $\epsilon < \frac{1}{2\sqrt{2}}$ satisfies $h_{\mathbf{w}}(\mathbf{x}) = -1$.*

- *There exists a choice of $\mathbf{w}$, such that if we run the algorithm on $h_{\mathbf{w}}(\cdot)$, then with probability at least $1 - T\exp(-d/18)$ over the algorithm's randomness (or deterministically if the algorithm is deterministic), the algorithm's iterates $\mathbf{x}_1, \ldots, \mathbf{x}_T$ satisfy $\min_{t \in \{1,\ldots,T\}} h_{\mathbf{w}}(\mathbf{x}_t) > 0$.*

*Proof.* The Lipschitz bound follows from the facts that $z \mapsto \max\{-1, z\}$ is 1-Lipschitz, $\mathbf{x} \mapsto M^{1/2}(\mathbf{x} - \mathbf{x}^*)$ is $\|M^{1/2}\| \leq 1$-Lipschitz, and that $g_{\mathbf{w}}$ is 7-Lipschitz by Lemma 2. Moreover, we clearly have $\inf_{\mathbf{x}} h_{\mathbf{w}}(\mathbf{x}) \geq -1$, and by definition of $h_{\mathbf{w}}(\cdot)$ and Lemma 1,

$$h_{\mathbf{w}}(\mathbf{0}) \leq \| - M^{1/2}\mathbf{x}^*\| = \hat{f}(\mathbf{0}) \leq \frac{1}{2} \, .$$

Combining the two observations establishes the first bullet in the lemma.

As to the second bullet, let $\hat{g}(\mathbf{x}) := g_{\mathbf{w}}(M^{1/2}(\mathbf{x} - \mathbf{x}^*))$ (so that $h_{\mathbf{w}}(\mathbf{x}) = \max\{-1, \hat{g}(\mathbf{x})\}$). It is easily verified that $\mathbf{u} \in \partial g_{\mathbf{w}}(\mathbf{x})$ if and only if $M^{1/2}\mathbf{u} \in \partial \hat{g}(\mathbf{x} + \mathbf{x}^*)$. By Lemma 2, $g_{\mathbf{w}}$ has no $\epsilon$-stationary point for $\epsilon < \frac{1}{\sqrt{2}}$, which implies that $\hat{g}(\mathbf{x})$ has no $\epsilon$-stationary points for any $\epsilon$ less than $\lambda_{\min}(M^{1/2})\frac{1}{\sqrt{2}} \geq \frac{1}{2\sqrt{2}}$. But since $h_{\mathbf{w}}(\mathbf{x}) = \max\{-1, \hat{g}(\mathbf{x})\}$, it follows that any $\epsilon$-stationary points of $h_{\mathbf{w}}(\cdot)$ must be arbitrarily close to the region where $h_{\mathbf{w}}(\cdot)$ is different than $\hat{g}(\cdot)$, namely where it takes a value of $-1$. Since $h_{\mathbf{w}}(\cdot)$ is Lipschitz, it follows that its value is $-1$ at the $\epsilon$-stationary point as well.

We now turn to establish the third bullet in the lemma. A crucial observation here is that

$$h_{\mathbf{w}}(\mathbf{x}) = g_{\mathbf{w}}(M^{1/2}(\mathbf{x} - \mathbf{x}^*)) = \hat{f}(\mathbf{x}) \quad \forall \mathbf{x} : \bar{\mathbf{w}}^\top \left( \overline{M^{1/2}(\mathbf{x} - \mathbf{x}^*) + \mathbf{w}} \right) \leq \frac{1}{2} \, , \qquad (7)$$

where $\hat{f}(\mathbf{x}) = \|M^{1/2}(\mathbf{x} - \mathbf{x}^*)\|$ is the "hard function" defined in Lemma 1[6]. To see why, note first that by definition of $g_{\mathbf{w}}(\cdot)$ in Lemma 2, for any $\mathbf{x}$ which satisfies the condition in the displayed equation above, we have $g_{\mathbf{w}}(M^{1/2}(\mathbf{x} - \mathbf{x}^*)) = \|M^{1/2}(\mathbf{x} - \mathbf{x}^*)\| = \hat{f}(\mathbf{x})$. On the other hand, since this is a non-negative function, it follows that it also equals $\max\{-1, g_{\mathbf{w}}(M^{1/2}(\mathbf{x} - \mathbf{x}^*))\} = h_{\mathbf{w}}(\mathbf{x})$ for such $\mathbf{x}$, establishing the displayed equation above.

Next, we will show that Eq. (7) also holds over a set of $\mathbf{x}$'s which have a more convenient form. To do so, fix some $\mathbf{x}$ which satisfies the *opposite* condition $\bar{\mathbf{w}}^\top \left( \overline{M^{1/2}(\mathbf{x} - \mathbf{x}^*) + \mathbf{w}} \right) > \frac{1}{2}$. Then multiplying both sides by $\|M^{1/2}(\mathbf{x} - \mathbf{x}^*) + \mathbf{w}\|$, we get

$$\bar{\mathbf{w}}^\top M^{1/2}(\mathbf{x} - \mathbf{x}^*) + \bar{\mathbf{w}}^\top \mathbf{w} > \frac{1}{2}\|M^{1/2}(\mathbf{x} - \mathbf{x}^*) + \mathbf{w}\| \geq \frac{1}{2}\left( \|M^{1/2}(\mathbf{x} - \mathbf{x}^*)\| - \|\mathbf{w}\| \right) \, .$$

Since $\frac{1}{2} \leq \lambda_{\min}(M) \leq \lambda_{\max}(M) \leq 1$ by Lemma 1, it follows that

$$\bar{\mathbf{w}}^\top M^{1/2}(\mathbf{x} - \mathbf{x}^*) + \|\mathbf{w}\| > \frac{1}{2}\left( \frac{1}{\sqrt{2}}\|\mathbf{x} - \mathbf{x}^*\| - \|\mathbf{w}\| \right) \, .$$

For $\mathbf{x} = \mathbf{x}^*$, the condition above is trivially satisfied. For $\mathbf{x} \neq \mathbf{x}^*$, dividing both sides by $\|M^{1/2}(\mathbf{x} - \mathbf{x}^*)\|$ (which is between $\|\mathbf{x} - \mathbf{x}^*\|$ and $\frac{1}{\sqrt{2}}\|\mathbf{x} - \mathbf{x}^*\|$) and simplifying a bit, we get that

$$\bar{\mathbf{w}}^\top \left( \overline{M^{1/2}(\mathbf{x} - \mathbf{x}^*)} \right) > \frac{1}{2\sqrt{2}} - \frac{3\|\mathbf{w}\|}{2 \cdot \frac{1}{\sqrt{2}}\|\mathbf{x} - \mathbf{x}^*\|} > \frac{1}{2\sqrt{2}} - \frac{\exp(-T)}{100\|\mathbf{x} - \mathbf{x}^*\|} \, .$$

Noting that any $\mathbf{x}$ which does not satisfy the condition in Eq. (7) satisfy the condition above, we get that Eq. (7) implies

$$h_{\mathbf{w}}(\mathbf{x}) = \hat{f}(\mathbf{x}) = \|M^{1/2}(\mathbf{x} - \mathbf{x}^*)\| \quad \forall \mathbf{x} \neq \mathbf{x}^* \text{ s.t. } \bar{\mathbf{w}}^\top \left( \overline{M^{1/2}(\mathbf{x} - \mathbf{x}^*)} \right) \leq \frac{1}{2\sqrt{2}} - \frac{\exp(-T)}{100\|\mathbf{x} - \mathbf{x}^*\|} \, . \qquad (8)$$

---

[6]Also, the equation can be verified to hold in the corner case where $M^{1/2}(\mathbf{x} - \mathbf{x}^*) + \mathbf{w} = \mathbf{0}$, in which the condition in Eq. (7) is undefined.

With this equation in hand, let us first establish the third bullet of the lemma, assuming the algorithm is in $\mathcal{A}_{det}$ (namely, it is deterministic). In order to do so, let $\mathbf{x}_1^{\hat{f}}, \ldots, \mathbf{x}_T^{\hat{f}}$ be the (fixed) iterates produced by the algorithm when ran on $\hat{f}(\cdot)$, and choose $\mathbf{w}$ in $h_{\mathbf{w}}(\cdot)$ to be any vector orthogonal to $\{M^{1/2}(\mathbf{x}_t^{\hat{f}} - \mathbf{x}^*)\}_{t=1}^T$ (which is possible since the dimension $d$ is larger than $T$). By Lemma 1, we know that for all $t$, $\|\mathbf{x}_t^{\hat{f}} - \mathbf{x}^*\| \geq \exp(-T)$, in which case we have

$$\bar{\mathbf{w}}^\top \left( \overline{M^{1/2}(\mathbf{x}_t^{\hat{f}} - \mathbf{x}^*)} \right) \;=\; 0 \;<\; \frac{1}{2\sqrt{2}} - \frac{\exp(-T)}{100\exp(-T)} \;\leq\; \frac{1}{2\sqrt{2}} - \frac{\exp(-T)}{100\|\mathbf{x}_t^{\hat{f}} - \mathbf{x}^*\|} \,.$$

Thus, $\mathbf{x}_t^{\hat{f}}$ satisfies the condition in Eq. (8), and as a result, $h_{\mathbf{w}}(\mathbf{x}_t^{\hat{f}}) = \hat{f}(\mathbf{x}_t^{\hat{f}})$ for all $t$. Moreover, using the fact that $\mathbf{x}_t^{\hat{f}}$ is bounded away from $\mathbf{x}^*$, it is easily verified that the condition in Eq. (8) also holds for $\mathbf{x}$ in a small local neighborhood of $\mathbf{x}_t^{\hat{f}}$, so actually $h_{\mathbf{w}}(\cdot)$ is identical to $\hat{f}(\cdot)$ on these local neigborhoods, implying the same values *and gradient sets* at $\mathbf{x}_t^{\hat{f}}$. As a result, if we run the algorithm on $h_{\mathbf{w}}(\cdot)$ rather than $f(\cdot)$, then the iterates $\mathbf{x}_1, \ldots, \mathbf{x}_T$ produced are identical to $\mathbf{x}_1^{\hat{f}}, \ldots, \mathbf{x}_T^{\hat{f}}$. Since $\|\mathbf{x}_t^{\hat{f}} - \mathbf{x}^*\| > 0$, we have $h_{\mathbf{w}}(\mathbf{x}_t) = \hat{f}(\mathbf{x}_t^{\hat{f}}) = \|M^{1/2}(\mathbf{x}_t^{\hat{f}} - \mathbf{x}^*)\| > 0$ for all $t$ as required.

We now turn to establish the third bullet of the lemma, assuming the algorithm is randomized. As before, we let $\mathbf{x}_1^{\hat{f}}, \ldots, \mathbf{x}_T^{\hat{f}}$ denote the iterates produced by the algorithm when ran on $\hat{f}(\cdot)$ (only that now they are possibly random, based on the algorithm's random coin flips). The proof idea is roughly the same, but here the iterates may be randomized, so we cannot choose $\mathbf{w}$ in some fixed manner. Instead, we will pick $\mathbf{w}$ independently and uniformly at random among vectors of norm $\frac{1}{300}\exp(-T)$, and show that for any realization of the algorithm's random coin flips, with probability at least $1 - T\exp(-d/18)$ over $\mathbf{w}$, $\min_t h_{\mathbf{w}}(\mathbf{x}_t) > 0$. This implies that there exists some *fixed* choice of $\mathbf{w}$, for which $\min_t h_{\mathbf{w}}(\mathbf{x}_t) > 0$ with the same high probability over the algorithm's randomness, as required[7]. To proceed, we collect two observations:

1. By Lemma 1, we know that for any realization of the algorithm's random coin flips, $\min_{t \in \{1, \ldots, T\}} \|\mathbf{x}_t^{\hat{f}} - \mathbf{x}^*\| \geq \exp(-T) > 0$.

2. If we fix some unit vectors $\mathbf{u}_1, \ldots, \mathbf{u}_T$ in $\mathbb{R}^d$, and pick a unit vector $\mathbf{u}$ uniformly at random, then by a union bound and a standard large deviation bound (e.g., [34]), $\Pr(\max_t \mathbf{u}^\top \mathbf{u}_t \geq a) \leq T \cdot \Pr(\mathbf{u}^\top \mathbf{u}_1 \geq a) \leq T\exp(-da^2/2)$. Taking $\overline{\mathbf{w}} = \mathbf{u}$, $\mathbf{u}_t = \overline{M^{1/2}(\mathbf{x}_t^{\hat{f}} - \mathbf{x}^*)}$ for all $t$ (for some realization of $\mathbf{x}_t^{\hat{f}}$), and $a = 1/3$, it follows that for any realization of the algorithm's random coin flips, $\max_t \overline{\mathbf{w}}^\top \overline{(M^{1/2}\mathbf{x}_t^{\hat{f}} - \mathbf{x}^*)} \geq 1/3$ with probability at most $T\exp(-d/18)$ over the choice of $\mathbf{w}$.

Combining the two observations, we get that for any realization of the algorithm's coin flips, with probability at least $1 - T\exp(-d/18)$ over the choice of $\mathbf{w}$, it holds for all $\mathbf{x}_1^{\hat{f}}, \ldots, \mathbf{x}_T^{\hat{f}}$ that

$$\bar{\mathbf{w}}^\top \left( \overline{M^{1/2}(\mathbf{x}_t^{\hat{f}} - \mathbf{x}^*)} \right) \;<\; \frac{1}{3} \;<\; \frac{1}{2\sqrt{2}} - \frac{\exp(-T)}{100\exp(-T)} \;\leq\; \frac{1}{2\sqrt{2}} - \frac{\exp(-T)}{100\|\mathbf{x}_t^{\hat{f}} - \mathbf{x}^*\|} \,,$$

as well as $\|\mathbf{x}_t^{\hat{f}} - \mathbf{x}^*\| > 0$. Using the same argument as in the deterministic case, it follows from Eq. (8) that $h_{\mathbf{w}}(\cdot)$ and $\hat{f}(\cdot)$ coincide in small neighborhoods around $\mathbf{x}_1^{\hat{f}}, \ldots, \mathbf{x}_T^{\hat{f}}$, with probability at least $1 - T\exp(-d/18)$. Since the algorithm's iterates depend only on the local values/gradient sets returned by the oracle, it follows that for any realization of the algorithm's coin flips, with probability at least $1 - T\exp(-d/18)$ over the choice of $\mathbf{w}$, the iterates $\mathbf{x}_1, \ldots, \mathbf{x}_T$ and $\mathbf{x}_1^{\hat{f}}, \ldots, \mathbf{x}_T^{\hat{f}}$ are going

---

[7]To see why, assume on the contrary that for any fixed choice of $\mathbf{w}$, the bad event $\min_t h_{\mathbf{w}}(\mathbf{x}_t) \leq 0$ occurs with probability larger than $T\exp(-d/18)$ over the algorithm's randomness. In that case, any randomization over the choice of $\mathbf{w}$ will still yield $\min_t h_{\mathbf{w}}(\mathbf{x}_t) \leq 0$ with probability larger than $T\exp(-d/18)$ over the joint randomness of $\mathbf{w}$ and the algorithm. In particular, this bad event will hold with probability larger than $T\exp(-d/18)$ for some realization of the algorithm's randomness.

to be identical, and satisfy

$$\min_t h_{\mathbf{w}}(\mathbf{x}_t) = \min_t h_{\mathbf{w}}(\mathbf{x}_t^{\hat{f}}) = \min_t \hat{f}(\mathbf{x}_t^{\hat{f}}) > 0 \, .$$

This holds for any realization of the algorithm's random coin flips, which as discussed earlier, implies the required result. □

The theorem is now an immediate corollary of the lemma above: With the specified high probability (or deterministically), $\min_t h_{\mathbf{w}}(\mathbf{x}_t) > 0$, even though all $\epsilon$-stationary points (for any $\epsilon < \frac{1}{2\sqrt{2}}$) have a value of $-1$. Since $h_{\mathbf{w}}$ is also 7-Lipschitz, we get that the distance of any $\mathbf{x}_t$ from an $\epsilon$-stationary point must be at least $\frac{0-(-1)}{7} = \frac{1}{7}$. Simplifying the numerical terms by choosing a large enough constant $C$ and a small enough constant $c$, and relabeling $h_{\mathbf{w}}$ as $f$, the theorem follows.

### A.2 Proof of Thm. 2

**Lemma 4.** *If $\mathcal{A}$ is an $(L, \epsilon, T, M, r)$-smoother satisfying TICF, then for any constant function $f$ and any $\mathbf{x} \in \mathbb{R}^d$: $\|\mathbb{E}[\mathcal{A}(f, \mathbf{x})]\| \leq \epsilon$.*

*Proof.* Denote $\mathbf{v} := \mathbb{E}[\mathcal{A}(f, \mathbf{x})]$, and note that by the TICF property $\mathbf{v}$ does not depend on $\mathbf{x}$. Let $\tilde{f}$ be the $\epsilon$-approximation of $f$ implicitly computed by $\mathcal{A}$, then by the definition of a smoothing algorithm, we have for all $\mathbf{x} \in \mathbb{R}^d$:

$$\left\| \mathbf{v} - \nabla \tilde{f}(\mathbf{x}) \right\| \leq \epsilon$$

$$\implies \|\mathbf{v}\|^2 - \left\langle \nabla \tilde{f}(\mathbf{x}), \mathbf{v} \right\rangle = \left\langle \mathbf{v} - \nabla \tilde{f}(\mathbf{x}), \mathbf{v} \right\rangle \leq \left\| \mathbf{v} - \nabla \tilde{f}(\mathbf{x}) \right\| \cdot \|\mathbf{v}\| \leq \epsilon \|\mathbf{v}\|$$

$$\implies \left\langle \nabla \tilde{f}(\mathbf{x}), \mathbf{v} \right\rangle \geq \|\mathbf{v}\|^2 - \epsilon \|\mathbf{v}\| \, .$$

Define the one dimensional projected function $\tilde{f}_{\mathbf{v}}(t) := \tilde{f}(t \cdot \mathbf{v})$. Then for all $t \geq 0$,

$$\tilde{f}_{\mathbf{v}}(t) - \tilde{f}_{\mathbf{v}}(0) = \int_0^t \tilde{f}_{\mathbf{v}}'(z) dz = \int_0^t \left\langle \nabla \tilde{f}(z \cdot \mathbf{v}), \mathbf{v} \right\rangle dz$$

$$\geq \int_0^t \left( \|\mathbf{v}\|^2 - \epsilon \|\mathbf{v}\| \right) dz = t \left( \|\mathbf{v}\|^2 - \epsilon \|\mathbf{v}\| \right) = t \|\mathbf{v}\| (\|\mathbf{v}\| - \epsilon) \, . \tag{9}$$

On the other hand, $\tilde{f}_{\mathbf{v}}(t), \tilde{f}_{\mathbf{v}}(0)$ are both $\epsilon$-approximations of the same constant, since $f$ is a constant function. Thus, $|\tilde{f}_{\mathbf{v}}(t) - \tilde{f}_{\mathbf{v}}(0)| \leq 2\epsilon$. Combining this with Eq. (9) yields for all $t \geq 0$

$$2\epsilon \geq t\|\mathbf{v}\|(\|\mathbf{v}\| - \epsilon) \tag{10}$$

This can hold for all $t \geq 0$ only if $(\|\mathbf{v}\| - \epsilon) \leq 0$, implying the lemma. □

We now show that without loss of generality we can impose certain assumptions on the parameters of interest. First, if $\epsilon \geq 1$ then the right hand side of Eq. (4) is negative for any $c_2 < 1$, which makes the theorem trivial. Consequently, we can assume $\epsilon < 1$. Using Lemma 11 in Appendix E, this also implies that $L \geq \frac{1}{8}$ since otherwise an $L$-smooth $\epsilon$-approximation does not exist in the first place in case of 1-Lipschitz function $\mathbf{x} \mapsto |x_1|$ (in particular, no such smoother exists). Therefore, if $\sqrt{\log((M+1)(T+1))} \geq \frac{\sqrt{d}}{32r}$ then

$$L\sqrt{\log((M+1)(T+1))} \geq \frac{1}{8} \cdot \frac{\sqrt{d}}{32r} > \frac{1}{256} \cdot \frac{\sqrt{d}}{r}(1 - \epsilon) \, ,$$

which proves the theorem. Thus we can assume throughout the proof that

$$\sqrt{\log((M+1)(T+1))} < \frac{\sqrt{d}}{32r} \implies \frac{1}{16r}\sqrt{\frac{d}{\log((M+1)(T+1))}} > 2 \, . \tag{11}$$

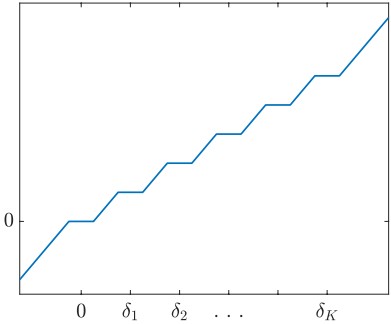

Figure 2: Illustration of $g(x)$, where $\Delta = \{0, \delta_1, \dots, \delta_K\} \subset [0,1]$

Our strategy is to define a distribution over a family of "hard" 1-Lipschitz functions over $\mathbb{R}^d$, for which we will show that Eq. (4) must hold for some function supported by this distribution. By Eq. (11) we can define the set

$$\Delta := \left\{ 16r\sqrt{\frac{\log\left(\left(M+1\right)\left(T+1\right)\right)}{d}} \cdot k \,\middle|\, k = 0, 1, \dots, \left\lfloor \frac{1}{16r}\sqrt{\frac{d}{\log\left(\left(M+1\right)\left(T+1\right)\right)}}\right\rfloor \right\}$$

That is, a grid on $[0,1]$ which consists of points of distance $16r\sqrt{\frac{\log((M+1)(T+1))}{d}}$ one from another. We further define the "inflation" of $\Delta$ by $4r\sqrt{\frac{\log((M+1)(T+1))}{d}}$ around every point:[8]

$$\overline{\Delta} := \left\{ x \in \mathbb{R} \,\middle|\, \exists p \in \Delta : |p - x| \le 4r\sqrt{\frac{\log\left(\left(M+1\right)\left(T+1\right)\right)}{d}} \right\}$$

Now we define the function $g : \mathbb{R} \to \mathbb{R}$ as the unique continuous function which satisfies (see Fig. 2 for an illustration)

$$g(0) = 0$$
$$g'(x) = \mathbf{1}_{\left\{x \notin \overline{\Delta}\right\}}$$

Finally, we are ready to consider

$$f_{\mathbf{w}}(\mathbf{x}) = g\left(\langle \mathbf{x}, \mathbf{w} \rangle\right) ,$$

where $\mathbf{w} \in \mathcal{S}^{d-1}$ is drawn uniformly from the unit sphere. The distribution over $\mathbf{w}$ specifies a distribution over the functions $f_{\mathbf{w}}$. We start by claiming that these functions are indeed in our function class of interest:

**Lemma 5.** *For all* $\mathbf{w} \in \mathcal{S}^{d-1}$, $f_{\mathbf{w}}(\cdot)$ *is 1-Lipschitz.*

*Proof.* It is clear by construction that $g$ is 1-Lipschitz. Thus

$$|f(\mathbf{x}) - f(\mathbf{y})| = |g\left(\langle \mathbf{x}, \mathbf{w} \rangle\right) - g\left(\langle \mathbf{y}, \mathbf{w} \rangle\right)| \le |\langle \mathbf{x}, \mathbf{w} \rangle - \langle \mathbf{y}, \mathbf{w} \rangle| = |\langle \mathbf{x} - \mathbf{y}, \mathbf{w} \rangle| \le \|\mathbf{x} - \mathbf{y}\|$$

$\square$

**Lemma 6.** *There exists* $\mathbf{w} \in \mathcal{S}^{d-1}$ *such that for all* $\delta \in \Delta$ : $\mathbb{E}_\xi\left[\|\mathcal{A}\left(f_{\mathbf{w}}, \delta\mathbf{w}\right)\|\right] \le \epsilon + \frac{1}{32}$.

*Proof.* Let $\mathbf{x}_1^{(\mathbf{w})}, \dots, \mathbf{x}_T^{(\mathbf{w})}$ be the (possibly randomized) queries produced by $\mathcal{A}\left(f_{\mathbf{w}}, \mathbf{0}\right)$. Fix some $\delta \in \Delta$, and let $\tilde{\mathbf{x}}_1^{(\mathbf{w})}, \dots, \tilde{\mathbf{x}}_T^{(\mathbf{w})}$ be the (possibly randomized) queries produced by $\mathcal{A}\left(f_{\mathbf{w}}, \delta\mathbf{w}\right)$. Consider the event $E_{\mathbf{w}}$, in which for all $i \in [T]$ : $\left|\langle \mathbf{x}_i^{(\mathbf{w})}, \mathbf{w} \rangle\right| < 4r\sqrt{\frac{\log((M+1)(T+1))}{d}}$. Note that if $E_{\mathbf{w}}$ occurs then for all $\delta \in \Delta, i \in [T], \mathbf{v} \in \mathbb{R}^d$:

$$f_{\mathbf{w}}\left(\mathbf{x}_i^{(\mathbf{w})} + \delta\mathbf{w} + \mathbf{v}\right) = g\left(\left\langle \mathbf{x}_i^{(\mathbf{w})} + \delta\mathbf{w} + \mathbf{v}, \mathbf{w} \right\rangle\right) = g\left(\delta + \left\langle \mathbf{x}_i^{(\mathbf{w})}, \mathbf{w} \right\rangle + \langle \mathbf{v}, \mathbf{w} \rangle\right) . \quad (12)$$

---

[8]Note we use the quantities $T+1, M+1$ instead of the seemingly more natural $T, M$, since otherwise the logarithmic term in Eq. (4) can vanish, resulting in an invalid theorem. This would have occurred for randomized smoothing, where $T = M = 1$.

In particular, as long as $\|\mathbf{v}\| < 4r\sqrt{\frac{\log((M+1)(T+1))}{d}} - \left|\left\langle \mathbf{x}_i^{(\mathbf{w})}, \mathbf{w}\right\rangle\right|$, which by Cauchy-Schwarz implies

$$\left|\left\langle \mathbf{x}_i^{(\mathbf{w})}, \mathbf{w}\right\rangle + \langle \mathbf{v}, \mathbf{w}\rangle\right| < 4r\sqrt{\frac{\log\left((M+1)\left(T+1\right)\right)}{d}},$$

we get by construction of $g$ and Eq. (12) that

$$f_{\mathbf{w}}\left(\mathbf{x}_i^{(\mathbf{w})} + \delta\mathbf{w} + \mathbf{v}\right) = g\left(\delta\right).$$

In other words, if $E_{\mathbf{w}}$ occurs then inside some neighborhood of $\mathbf{x}_i^{(\mathbf{w})} + \delta\mathbf{w}$, the function $f_{\mathbf{w}}$ is identical to the constant function $g\left(\delta\right)$. Therefore, if $E_{\mathbf{w}}$ occurs the all-derivatives oracle $\mathsf{O}^\infty$ satisfies

$$\mathsf{O}_{f_{\mathbf{w}}}^\infty\left(\mathbf{x}_i^{(\mathbf{w})} + \delta\mathbf{w}\right) = \mathsf{O}_{\mathbf{x}\mapsto g(\delta)}^\infty. \tag{13}$$

Recall that we use the abbreviation $\mathsf{O}_{\mathbf{x}\mapsto g(\delta)}^\infty$ since the oracle's response does not depend on the input point for constant functions. We will now show that conditioned on $E_{\mathbf{w}}$, for all $i \in [T]$:

$$\tilde{\mathbf{x}}_i^{(\mathbf{w})} = \mathbf{x}_i^{(\mathbf{w})} + \delta\mathbf{w}, \tag{14}$$

in the sense that for every realization of $\mathcal{A}'$s randomness $\xi$ they are equal. We show this by induction on $i$. For $i = 1$, using TICF:

$$\tilde{\mathbf{x}}_1^{(\mathbf{w})} = \mathcal{A}^{(1)}\left(\xi, \delta\mathbf{w}\right) = \mathcal{A}^{(1)}\left(\xi, \mathbf{0}\right) + \delta\mathbf{w} = \mathbf{x}_1^{(\mathbf{w})} + \delta\mathbf{w}.$$

Assuming this is true up until $i$, then by the induction hypothesis, Eq. (13) and TICF:

$$\begin{aligned}
\tilde{\mathbf{x}}_{i+1}^{(\mathbf{w})} &= \mathcal{A}^{(i)}\left(\xi, \delta\mathbf{w}, \mathsf{O}_{f_{\mathbf{w}}}^\infty\left(\tilde{\mathbf{x}}_1^{(\mathbf{w})}\right), \ldots, \mathsf{O}_{f_{\mathbf{w}}}^\infty\left(\tilde{\mathbf{x}}_i^{(\mathbf{w})}\right)\right) \\
&= \mathcal{A}^{(i)}\left(\xi, \delta\mathbf{w}, \mathsf{O}_{f_{\mathbf{w}}}^\infty\left(\mathbf{x}_1^{(\mathbf{w})} + \delta\mathbf{w}\right), \ldots, \mathsf{O}_{f_{\mathbf{w}}}^\infty\left(\mathbf{x}_i^{(\mathbf{w})} + \delta\mathbf{w}\right)\right) \\
&= \mathcal{A}^{(i)}\left(\xi, \delta\mathbf{w}, \mathsf{O}_{\mathbf{x}\mapsto g(\delta)}^\infty, \ldots, \mathsf{O}_{\mathbf{x}\mapsto g(\delta)}^\infty\right) \\
&= \mathcal{A}^{(i)}\left(\xi, \mathbf{0}, \mathsf{O}_{\mathbf{x}\mapsto g(\delta)}^\infty, \ldots, \mathsf{O}_{\mathbf{x}\mapsto g(\delta)}^\infty\right) + \delta\mathbf{w} \\
&= \mathcal{A}^{(i)}\left(\xi, \mathbf{0}, \mathsf{O}_{f_{\mathbf{w}}}^\infty\left(\mathbf{x}_1^{(\mathbf{w})}\right), \ldots, \mathsf{O}_{f_{\mathbf{w}}}^\infty\left(\mathbf{x}_i^{(\mathbf{w})}\right)\right) + \delta\mathbf{w} \\
&= \mathbf{x}_{i+1}^{(\mathbf{w})} + \delta\mathbf{w}.
\end{aligned}$$

Having established Eq. (14), we turn to show that for all $\delta \in \Delta$:

$$\mathbb{E}_\xi\left[\mathcal{A}\left(f_{\mathbf{w}}, \delta\mathbf{w}\right)|E_{\mathbf{w}}\right] = \mathbb{E}_\xi\left[\mathcal{A}\left(\mathbf{x}\mapsto 0, \mathbf{0}\right)|E_{\mathbf{w}}\right]. \tag{15}$$

Indeed, by Eq. (14), Eq. (13) and TICF:

$$\begin{aligned}
\mathbb{E}_\xi\left[\mathcal{A}\left(f_{\mathbf{w}}, \delta\mathbf{w}\right)|E_{\mathbf{w}}\right] &= \mathbb{E}_\xi\left[\mathcal{A}^{(out)}\left(\xi, \delta\mathbf{w}, \mathsf{O}_{f_{\mathbf{w}}}^\infty\left(\tilde{\mathbf{x}}_1^{(\mathbf{w})}\right), \ldots, \mathsf{O}_{f_{\mathbf{w}}}^\infty\left(\tilde{\mathbf{x}}_T^{(\mathbf{w})}\right)\right)\Big|E_{\mathbf{w}}\right] \\
&= \mathbb{E}_\xi\left[\mathcal{A}^{(out)}\left(\xi, \delta\mathbf{w}, \mathsf{O}_{f_{\mathbf{w}}}^\infty\left(\mathbf{x}_1^{(\mathbf{w})} + \delta\mathbf{w}\right), \ldots, \mathsf{O}_{f_{\mathbf{w}}}^\infty\left(\mathbf{x}_T^{(\mathbf{w})} + \delta\mathbf{w}\right)\right)\Big|E_{\mathbf{w}}\right] \\
&= \mathbb{E}_\xi\left[\mathcal{A}^{(out)}\left(\xi, \delta\mathbf{w}, \mathsf{O}_{\mathbf{x}\mapsto g(\delta)}^\infty, \ldots, \mathsf{O}_{\mathbf{x}\mapsto g(\delta)}^\infty\right)\Big|E_{\mathbf{w}}\right] \\
&= \mathbb{E}_\xi\left[\mathcal{A}^{(out)}\left(\xi, \mathbf{0}, \mathsf{O}_{\mathbf{x}\mapsto 0}^\infty, \ldots, \mathsf{O}_{\mathbf{x}\mapsto 0}^\infty\right)\Big|E_{\mathbf{w}}\right] \\
&= \mathbb{E}_\xi\left[\mathcal{A}\left(\mathbf{x}\mapsto 0, \mathbf{0}\right)|E_{\mathbf{w}}\right].
\end{aligned}$$

We now turn to show that $E_{\mathbf{w}}$ is likely to occur. Fix some realization of $\mathcal{A}$'s randomness $\xi$, and let $\mathbf{q}_1^\xi, \ldots, \mathbf{q}_T^\xi$ be the (deterministic) queries produced by $\mathcal{A}\left(\mathbf{y}\mapsto 0, \mathbf{0}\right)$. We claim that if for all $i \in [T]: \left|\langle \mathbf{q}_i^\xi, \mathbf{w}\rangle\right| < 4r\sqrt{\frac{\log((M+1)(T+1))}{d}}$ then $\left(\mathbf{q}_1^\xi, \ldots, \mathbf{q}_T^\xi\right) = \left(\mathbf{x}_1^{(\mathbf{w})}, \ldots, \mathbf{x}_T^{(\mathbf{w})}\right)$ independently of $\mathbf{w}$. We show this by induction on $i$. For $i = 1$:

$$\mathbf{q}_1^\xi = \mathcal{A}^{(1)}\left(\xi, \mathbf{0}\right) = \mathbf{x}_1^{(\mathbf{w})}.$$

Assuming true up until $i$, then

$$
\begin{aligned}
\mathbf{q}_{i+1}^{\xi} &= \mathcal{A}^{(i)}\left(\xi, \mathbf{0}, \mathsf{O}_{\mathbf{x} \mapsto 0}^{\infty}\left(\mathbf{q}_1^{\xi}\right), \ldots, \mathsf{O}_{\mathbf{x} \mapsto 0}^{\infty}\left(\mathbf{q}_i^{\xi}\right)\right) \\
&= \mathcal{A}^{(i)}\left(\xi, \mathbf{0}, \mathsf{O}_{f_{\mathbf{w}}}^{\infty}\left(\mathbf{q}_1^{\xi}\right), \ldots, \mathsf{O}_{f_{\mathbf{w}}}^{\infty}\left(\mathbf{q}_i^{\xi}\right)\right) \\
&= \mathcal{A}^{(i)}\left(\xi, \mathbf{0}, \mathsf{O}_{f_{\mathbf{w}}}^{\infty}\left(\mathbf{x}_1^{(\mathbf{w})}\right), \ldots, \mathsf{O}_{f_{\mathbf{w}}}^{\infty}\left(\mathbf{x}_i^{(\mathbf{w})}\right)\right) \\
&= \mathbf{x}_{i+1}^{(\mathbf{w})},
\end{aligned}
$$

where we used the assumption on $\mathbf{q}_i^{\xi}$ and the induction hypothesis. Recall that by assumption on the algorithm $\left\|\mathbf{q}_i^{\xi}\right\| \leq r$ for all $i \in [T]$. Using the union bound and concentration of measure on the sphere (e.g., [34]) we get

$$
\begin{aligned}
\Pr_{\mathbf{w}}\left[\neg E_{\mathbf{w}} \mid \xi\right] &= \Pr_{\mathbf{w}}\left[\exists i \in [T]: \left|\langle \mathbf{q}_i^{\xi}, \mathbf{w} \rangle\right| \geq 4r\sqrt{\frac{\log\left((M+1)\,(T+1)\right)}{d}}\right] \\
&= \Pr_{\mathbf{w}}\left[\exists i \in [T]: \left|\left\langle \frac{1}{r}\mathbf{q}_i^{\xi}, \mathbf{w} \right\rangle\right| \geq 4\sqrt{\frac{\log\left((M+1)\,(T+1)\right)}{d}}\right] \\
&\leq T \cdot 2\exp\left(-\frac{d \cdot \left(4\sqrt{\frac{\log((M+1)(T+1))}{d}}\right)^2}{2}\right) \\
&= \frac{2T}{(M+1)^8\,(T+1)^8} \leq \frac{2}{(M+1)^8\,(T+1)^7}.
\end{aligned}
$$

This inequality holds for any realization of $\mathcal{A}$'s randomness $\xi$, hence by the law of total probability

$$
\Pr_{\xi, \mathbf{w}}\left[\neg E_{\mathbf{w}}\right] \leq \frac{2}{(M+1)^8\,(T+1)^7}.
$$

In particular, since $\Pr_{\xi, \mathbf{w}}\left[\neg E_{\mathbf{w}}\right] = \mathbb{E}_{\mathbf{w}}\left[\Pr_{\xi}\left[\neg E_{\mathbf{w}} | \mathbf{w}\right]\right]$, there exists $\mathbf{w} \in \mathcal{S}^{d-1}$ such that

$$
\Pr_{\xi}\left[\neg E_{\mathbf{w}}\right] \leq \frac{2}{(M+1)^8\,(T+1)^7}. \tag{16}
$$

For this fixed $\mathbf{w}$, we have for all $\delta \in \Delta$ by the law of total expectation and the triangle inequality:

$$
\left\|\mathbb{E}_{\xi}\left[\mathcal{A}\left(f_{\mathbf{w}}, \delta\mathbf{w}\right)\right]\right\| \leq \underbrace{\left\|\mathbb{E}_{\xi}\left[\mathcal{A}\left(f_{\mathbf{w}}, \delta\mathbf{w}\right) | E_{\mathbf{w}}\right] \cdot \Pr_{\xi}\left[E_{\mathbf{w}}\right]\right\|}_{(*)} + \underbrace{\left\|\mathbb{E}_{\xi}\left[\mathcal{A}\left(f_{\mathbf{w}}, \delta\mathbf{w}\right) | \neg E_{\mathbf{w}}\right] \cdot \Pr_{\xi}\left[\neg E_{\mathbf{w}}\right]\right\|}_{(**)}
$$

$$\tag{17}$$

On one hand, by Eq. (15):

$$
(*) = \mathbb{E}_{\xi}\left[\mathcal{A}\left(\mathbf{x} \mapsto 0, \mathbf{0}\right) | E_{\mathbf{w}}\right] \cdot \Pr_{\xi}\left[E_{\mathbf{w}}\right] = \mathbb{E}_{\xi}\left[\mathcal{A}\left(\mathbf{x} \mapsto 0, \mathbf{0}\right)\right] - \mathbb{E}_{\xi}\left[\mathcal{A}\left(\mathbf{x} \mapsto 0, \mathbf{0}\right) | \neg E_{\mathbf{w}}\right] \cdot \Pr_{\xi}\left[\neg E_{\mathbf{w}}\right]
$$

Using Lemma 4, and by incorporating the definition of $M$ in Eq. (2) and Eq. (16) we get

$$
\|(*)\| \leq \epsilon + M \cdot \frac{2}{(M+1)^8\,(T+1)^7} \leq \epsilon + \frac{2}{(M+1)^7\,(T+1)^7}. \tag{18}
$$

On the other hand, by Eq. (2) and Eq. (16) again we have

$$
\|(**)\| \leq \left\|\mathbb{E}_{\xi}\left[\mathcal{A}\left(f_{\mathbf{w}}, \delta\mathbf{w}\right) | \neg E_{\mathbf{w}}\right]\right\| \cdot \Pr_{\xi}\left[\neg E_{\mathbf{w}}\right] \leq M \cdot \frac{2}{(M+1)^8\,(T+1)^7} \leq \frac{2}{(M+1)^7\,(T+1)^7}. \tag{19}
$$

Overall, plugging Eq. (18) and Eq. (19) into Eq. (17), gives

$$
\left\|\mathbb{E}_{\xi}\left[\mathcal{A}\left(f_{\mathbf{w}}, \delta\mathbf{w}\right)\right]\right\| \leq \epsilon + \frac{4}{(M+1)^7\,(T+1)^7} \leq \epsilon + \frac{1}{32},
$$

where the last inequality simply follows from the fact that $M > 0$, $T \geq 1$. $\qquad \square$

From now on, we fix $\mathbf{w} \in \mathcal{S}^{d-1}$ which is given by the previous lemma and denote $f = f_{\mathbf{w}}$. Denote by $\tilde{f}$ the $\epsilon$-approximation of $f$ with $L$-Lipschitz gradients implicitly computed by $\mathcal{A}$. We turn our focus to the directional projection:

$$\varphi : [0,1] \to \mathbb{R}$$
$$\varphi(t) = \tilde{f}(t \cdot \mathbf{w})$$

Note that by assumption on $\tilde{f}$, $\varphi$ is differentiable, and $\varphi'$ is $L$-Lipschitz. Lemma 6 ensures us that $\varphi'$ is relatively close to zero on the grid $\Delta$, as showed in the following lemma.

**Lemma 7.** $\forall \delta \in \Delta : |\varphi'(\delta)| \le 2\epsilon + \frac{1}{32}$

*Proof.* By Cauchy-Schwarz, Lemma 6 and the definition of a smoother, we get that for all $\delta \in \Delta$:

$$|\varphi'(\delta)| = \left| \left\langle \nabla \tilde{f}(\delta \mathbf{w}), \mathbf{w} \right\rangle \right| \le \left\| \nabla \tilde{f}(\delta \mathbf{w}) \right\| \cdot \|\mathbf{w}\| = \left\| \nabla \tilde{f}(\delta \mathbf{w}) \right\|$$

$$\le \left\| \mathbb{E}[\mathcal{A}(f, \delta \mathbf{w})] - \nabla \tilde{f}(\delta \mathbf{w}) \right\| + \|\mathbb{E}[\mathcal{A}(f, \delta \mathbf{w})]\| \le \epsilon + \epsilon + \frac{1}{32}$$

$\square$

By combining the fact that $\varphi'$ has small values along the grid $\Delta$, with the fact that $\varphi'$ is $L$-Lipschitz, we can bound the oscillation of $\varphi$ along the unit interval.

**Lemma 8.** $|\varphi(1) - \varphi(0)| \le 2\epsilon + \frac{1}{32} + \frac{4Lr\sqrt{\log((M+1)(T+1))}}{\sqrt{d}}$

*Proof.* Denote $\delta_i = 16r\sqrt{\frac{\log((M+1)(T+1))}{d}} \cdot i$, and note that for all $i \in \left[ \left\lfloor \frac{1}{16r} \sqrt{\frac{d}{\log((M+1)(T+1))}} \right\rfloor \right]$:
$\delta_i \in \Delta$. Then

$$|\varphi(1) - \varphi(0)| = \left| \int_0^1 \varphi'(t)\,dt \right| \le \int_0^1 |\varphi'(t)|\,dt = \sum_{i=0}^{\left\lfloor \frac{1}{16r} \sqrt{\frac{d}{\log((M+1)(T+1))}} \right\rfloor - 1} \int_{\delta_i}^{\delta_{i+1}} |\varphi'(t)|\,dt$$

$$\le \left( \frac{1}{16r} \sqrt{\frac{d}{\log((M+1)(T+1))}} \right) \cdot \max_i \int_{\delta_i}^{\delta_{i+1}} |\varphi'(t)|\,dt \tag{20}$$

By Lemma 7 we have $|\varphi'(\delta_i)|, |\varphi'(\delta_{i+1})| \le 2\epsilon + \frac{1}{32}$. Recall that $\varphi'$ is $L$-Lipschitz, so $|\varphi'(t)|$ is majorized on the interval $[\delta_i, \delta_{i+1}]$ by the piecewise linear function (see Fig. 3)

$$l(t) = \begin{cases} 2\epsilon + \frac{1}{32} + L(t - \delta_i) & \delta_i \le t \le \frac{\delta_i + \delta_{i+1}}{2} \\ 2\epsilon + \frac{1}{32} + L(\delta_{i+1} - t) & \frac{\delta_i + \delta_{i+1}}{2} < t \le \delta_{i+1} \end{cases}$$

Consequently,

$$\int_{\delta_i}^{\delta_{i+1}} |\varphi'(t)|\,dt \le \int_{\delta_i}^{\delta_{i+1}} l(t)\,dt$$

$$= \left( 2\epsilon + \frac{1}{32} \right) \cdot 16r\sqrt{\frac{\log((M+1)(T+1))}{d}} + L\left( 8r\sqrt{\frac{\log((M+1)(T+1))}{d}} \right)^2, \tag{21}$$

where the last equality is a direct calculation. Plugging Eq. (21) into Eq. (20), we get that

$$|\varphi(1) - \varphi(0)| \le 2\epsilon + \frac{1}{32} + \frac{4Lr\sqrt{\log((M+1)(T+1))}}{\sqrt{d}}$$

$\square$

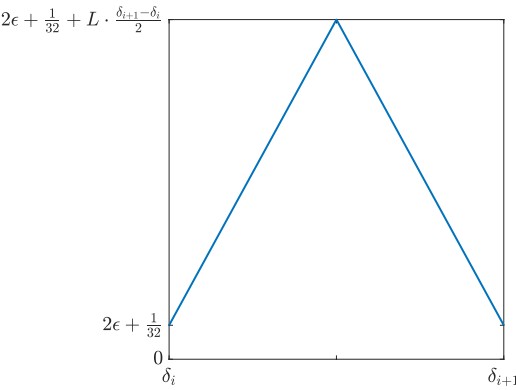

Figure 3: Illustration of $l(t)$

We are now ready to finish the proof. Notice that $\varphi(0) = \tilde{f}(0)$, $\varphi(1) = \tilde{f}(\mathbf{w})$. Additionally, a direct calculation shows that $f(\mathbf{0}) = 0$, $f(\mathbf{w}) \geq \frac{1}{2}$. Using the fact that $\|\tilde{f} - f\|_\infty \leq \epsilon$, Lemma 8 reveals

$$\frac{1}{2} \leq |f(\mathbf{w}) - f(0)| \leq \left|\tilde{f}(\mathbf{w}) - \tilde{f}(0)\right| + 2\epsilon = |\varphi(1) - \varphi(0)| + 2\epsilon$$

$$\leq 4\epsilon + \frac{1}{32} + \frac{4Lr\sqrt{\log((M+1)(T+1))}}{\sqrt{d}}$$

$$\implies L\sqrt{\log((M+1)(T+1))} \geq \frac{1}{16} \cdot \frac{\sqrt{d}}{r}\left(\frac{15}{128} - \epsilon\right).$$

## B   $(\delta, \epsilon)$-Stationarity

In the recent work by Zhang, Lin, Sra and Jadbabaie [37], the authors prove that for nonconvex nonsmooth functions, finding $\epsilon$-approximately stationary points is infeasible in general. Instead, they study the following relaxation (based on the notion of $\delta$-differential introduced in [21]): Letting $\partial f(\mathbf{x})$ denote the generalized gradient set[9] of $f(\cdot)$ at $\mathbf{x}$, we say that a point $\mathbf{x}$ is a $(\delta, \epsilon)$-stationary point, if

$$\min\{\|\mathbf{u}\| : \mathbf{u} \in \text{conv}\{\cup_{\mathbf{y}:\|\mathbf{y}-\mathbf{x}\|\leq\delta} \partial f(\mathbf{y})\}\} \leq \epsilon, \tag{22}$$

where $\text{conv}\{\cdot\}$ is the convex hull. In words, there exists a convex combination of gradients at a $\delta$-neighborhood of $\mathbf{x}$, whose norm is at most $\epsilon$. Remarkably, the authors then proceed to provide a dimension-free, gradient-based algorithm for finding $(\delta, \epsilon)$-stationary points, using $\mathcal{O}(1/\delta\epsilon^3)$ gradient and value evaluations, as well as study related settings.

Although this constitutes a very useful algorithmic contribution to nonsmooth optimization, it is important to note that a $(\delta, \epsilon)$-stationary point $\mathbf{x}$ (as defined above) *does not* imply that $\mathbf{x}$ is $\delta$-close to an $\epsilon$-stationary point of $f(\cdot)$, nor that $\mathbf{x}$ necessarily resembles a stationary point. Intuitively, this is because the convex hull of the gradients might contain a small vector, without any of the gradients being particular small. This is formally demonstrated in the following proposition:

**Proposition 1.** *For any $\delta > 0$, there exists a differentiable function $f(\cdot)$ on $\mathbb{R}^2$ which is $2\pi$-Lipschitz on a ball of radius $2\delta$ around the origin, and the origin is a $(\delta, 0)$-stationary point, yet $\min_{\mathbf{x}:\|\mathbf{x}\|\leq\delta} \|\nabla f(\mathbf{x})\| \geq 1$.*

*Proof.* Fixing some $\delta > 0$, consider the function

$$f(u, v) := (2\delta + u)\sin\left(\frac{\pi}{2\delta}v\right)$$

(see Fig. B for an illustration). This function is differentiable, and its gradient satisfies

$$\nabla f(u, v) = \left(\sin\left(\frac{\pi}{2\delta}v\right), \frac{\pi}{2\delta}(2\delta + u)\cos\left(\frac{\pi}{2\delta}v\right)\right).$$

---

[9]See Sec. 2 for the formal definition.

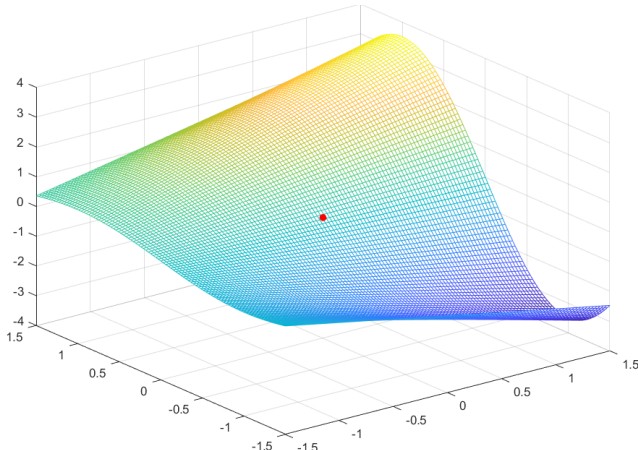

Figure 4: The function used in the proof of Proposition 1, for $\delta = 1$. The origin (which fulfills the definition of a $(1,0)$-stationary point) is marked with a red dot. Best viewed in color.

First, we note that

$$\frac{1}{2}\left(\nabla f(0,\delta) + \frac{1}{2}\nabla f(0,-\delta)\right) = \frac{1}{2}\left((1,0) + (-1,0)\right) = (0,0),$$

which implies that $(0,0)$ is in the convex hull of the gradients at a distance at most $\delta$ from the origin, hence the origin is a $(\delta, 0)$-stationary point. Second, we have that

$$\|\nabla f(u,v)\|^2 = \sin^2\left(\frac{\pi}{2\delta}v\right) + \left(\frac{\pi}{2\delta}\right)^2 (2\delta + u)^2 \cos^2\left(\frac{\pi}{2\delta}v\right). \tag{23}$$

For any $(u,v)$ of norm at most $2\delta$, we must have $|u| \leq 2\delta$, and therefore the above is at most

$$\sin^2\left(\frac{\pi}{2\delta}v\right) + \left(\frac{\pi}{2\delta}\right)^2 (2\delta + 2\delta)^2 \cos^2\left(\frac{\pi}{2\delta}v\right) \leq 4\pi^2\left(\sin^2\left(\frac{\pi}{2\delta}v\right) + \cos^2\left(\frac{\pi}{2\delta}v\right)\right) = 4\pi^2,$$

which implies that the function is $2\pi$-Lipschitz on a ball of radius $2\delta$ around the origin. Finally, for any $(u,v)$ of norm at most $\delta$, we have $|u| \leq \delta$, so Eq. (23) is at least

$$\sin^2\left(\frac{\pi}{2\delta}v\right) + \left(\frac{\pi}{2\delta}\right)^2 (2\delta - \delta)^2 \cos^2\left(\frac{\pi}{2\delta}v\right) \geq \sin^2\left(\frac{\pi}{2\delta}v\right) + \cos^2\left(\frac{\pi}{2\delta}v\right) = 1.$$

$\square$

**Remark 6** (Extension to globally Lipschitz functions). *Although the function $f(\cdot)$ in the proof has a constant Lipschitz parameter only close to the origin, it can be easily modified to be globally Lipschitz and bounded, for example by considering the function*

$$\tilde{f}(\mathbf{x}) = \begin{cases} f(\mathbf{x}) & \|\mathbf{x}\| \leq 2\delta \\ \max\left\{0, 2 - \frac{\|\mathbf{x}\|}{2\delta}\right\} \cdot f\left(\frac{2\delta}{\|\mathbf{x}\|}\mathbf{x}\right) & \|\mathbf{x}\| > 2\delta \end{cases},$$

*which is identical to $f(\cdot)$ in a ball of radius $2\delta$ around the origin, but decays to $0$ for larger $\mathbf{x}$, and can be verified to be globally bounded and Lipschitz independent of $\delta$.*

**Remark 7** (Extension to constant distances). *The proof of Thm. 1 uses a (more complicated) construction that actually strengthens Proposition 1: It implies that for any $\delta, \epsilon$ smaller than some constants, there is a Lipschitz, bounded-from-below function on $\mathbb{R}^d$, such that the origin is $(\delta, 0)$-stationary, yet there are no $\epsilon$-stationary points even at a constant distance from the origin. In more details, consider the function*

$$\hat{g}_{\mathbf{w}}(\mathbf{x}) := \max\{g_{\mathbf{w}}(\mathbf{0}) - 1, g_{\mathbf{w}}(\mathbf{x})\},$$

*where $g_{\mathbf{w}}(\cdot)$ is as defined in Lemma 2. $g_{\mathbf{w}}(\cdot)$ is 7-Lipschitz and has no $\epsilon$-stationary points for $\epsilon < 1/\sqrt{2}$. Therefore, it is easily verified that for any $\mathbf{w}$, $\hat{g}_{\mathbf{w}}(\cdot)$ is 7-Lipschitz, bounded from below,*

*and any $\epsilon$-stationary point is at a distance of at least $1/7$ from the origin[10]. However, we also claim that the origin is a $(\delta, 0)$-stationary point for any $\delta \in (0, 1/7)$. To see this, note first that for such $\delta$, by the Lipschitz property of $g_{\mathbf{w}}(\cdot)$, we have $\hat{g}_{\mathbf{w}}(\mathbf{x}) = g_{\mathbf{w}}(\mathbf{x})$ in a $\delta$-neighborhood of the origin. Fix any $\mathbf{w}$ such that $\|\mathbf{w}\| = \frac{\delta}{2}$, and let $\mathbf{v}$ be any vector of norm $\delta$ orthogonal to $\mathbf{w}$. It is easily verified that $\bar{\mathbf{w}}^\top (\mathbf{v} + \mathbf{w}) < \frac{1}{2}$, in which case*

$$\nabla \hat{g}_{\mathbf{w}}(\mathbf{v}) = \nabla g_{\mathbf{w}}(\mathbf{v}) = \bar{\mathbf{v}},$$

*and therefore $\frac{1}{2} (\nabla \hat{g}_{\mathbf{w}}(\mathbf{v}) + \nabla \hat{g}_{\mathbf{w}}(-\mathbf{v})) = \mathbf{0}$.*

We end by noting that if we drop the the conv$\{\cdot\}$ operator from the definition of $(\delta, \epsilon)$-stationarity in Eq. (22), the goal becomes equivalent to finding points which are $\delta$-close to $\epsilon$-approximately stationary points – which is exactly the goal we study in Sec. 3, and for which we show a strong impossibility result. This impossibility result implies that a natural strengthening of the notion of $(\delta, \epsilon)$-stationarity is already too strong to be feasible in general.

## C   Smoothed GD suffers from dimension

In this appendix, we formally prove that randomized smoothing can indeed lead to strong dimension dependencies in the iteration complexity of simple gradient methods – in particular, vanilla gradient descent with constant step size – even for simple convex functions. Thus, the dimension dependency arising from applying gradient descent on a randomly-smoothed function is real and not merely an artifact of the analysis (where the standard upper bound on the number of iterations scales with the gradient Lipschitz parameter). We note that we focus on constant step-size gradient descent for simplicity, and a similar analysis can be performed for other gradient-based methods, such as variable step-size gradient descent or stochastic gradient descent.

Given a 1-Lipschitz function $f : \mathbb{R}^d \to \mathbb{R}$, denote the smooth approximation $\tilde{f}(\mathbf{x}) = \mathbb{E}_{\|\mathbf{v}\| \leq 1}[f(\mathbf{x} + \epsilon \mathbf{v})]$ where $\mathbf{v}$ is distributed uniformly over the unit ball. Let $\mathbf{x}_0$ be a point which is of distance at most 1 to an $\epsilon$-stationary point of $\tilde{f}$, and consider vanilla gradient descent with a constant step size $\eta > 0$:

$$\mathbf{x}_{t+1} = \mathbf{x}_t - \eta \cdot \nabla \tilde{f}(\mathbf{x}_t) .$$

The following proposition shows that for any step size, applying gradient descent to find an approximately-stationary point of $\tilde{f}$ will necessitate a number of iterations scaling strongly with the dimension:

**Proposition 2.** *For any $\epsilon < \frac{1}{2}$, $\eta > 0$, there exists $f, \mathbf{x}_0$ as above such that $\min\{t : \|\nabla \tilde{f}(\mathbf{x}_t)\| \leq \epsilon\} = \Omega\left(\frac{\sqrt{d}}{\epsilon}\right)$.*

*Proof.* We will show the claim holds for $f(\mathbf{x}) := |x_1|$. In a nutshell, the proof is based on the observation that $\nabla \tilde{f}(\mathbf{x})$ is close to zero only when $|x_1| = \mathcal{O}(1/\sqrt{d})$. Thus, gradient descent must hit an interval of size $\mathcal{O}(1/\sqrt{d})$. But in order to guarantee this, and with an arbitrary bounded starting point, the step size must be small, and hence the number of iterations required will be large.

Proceeding with the formal proof, note that $\tilde{f}(\mathbf{x}) = \mathbb{E}_{\|\mathbf{v}\| \leq 1} [|x_1 + \epsilon v_1|]$, hence

$$
\begin{aligned}
\nabla \tilde{f}(\mathbf{x}) &= \mathbb{E}_{\|\mathbf{v}\| \leq 1} [\text{sign}(x_1 + \epsilon v_1)] \cdot \mathbf{e}_1 \\
&= \left( \Pr_{\|\mathbf{v}\| \leq 1} [x_1 + \epsilon v_1 > 0] - \Pr_{\|\mathbf{v}\| \leq 1} [x_1 + \epsilon v_1 < 0] \right) \cdot \mathbf{e}_1 \\
&= \left( 1 - 2 \cdot \Pr_{\|\mathbf{v}\| \leq 1} [x_1 + \epsilon v_1 < 0] \right) \cdot \mathbf{e}_1 \\
&= \left( 1 - 2 \cdot \Pr_{\|\mathbf{v}\| \leq 1} \left[ v_1 < -\frac{x_1}{\epsilon} \right] \right) \cdot \mathbf{e}_1
\end{aligned}
\tag{24}
$$

---

[10]The last point follows from the fact that if $\mathbf{y}$ is an $\epsilon$-stationary point of $\hat{g}_{\mathbf{w}}(\cdot)$, then we can find a point $\mathbf{x}$ arbitrarily close to $\mathbf{y}$ such that $\hat{g}_{\mathbf{w}}(\mathbf{x}) \neq g_{\mathbf{w}}(\mathbf{x})$, hence $g_{\mathbf{w}}(\mathbf{x}) < g_{\mathbf{w}}(\mathbf{0}) - 1$, and as a result $g_{\mathbf{w}}(\mathbf{0}) - g_{\mathbf{w}}(\mathbf{x}) > 1$. But $g_{\mathbf{w}}(\cdot)$ is 7-Lipschitz, hence $\|\mathbf{x}\| > 1/7$, and therefore $\|\mathbf{y}\| \geq 1/7$.

We draw several consequences from Eq. (24). First, if $x_1 = 0$ then $\Pr_{\|\mathbf{v}\| \leq 1}\left[v_1 < -\frac{x_1}{\epsilon}\right] = \frac{1}{2}$ due to symmetry around the origin, so in particular

$$\nabla \tilde{f}(\mathbf{0}) = \mathbf{0} . \tag{25}$$

Second, if $x_1 \geq \epsilon$ then $\Pr_{\|\mathbf{v}\| \leq 1}\left[v_1 < -\frac{x_1}{\epsilon}\right] = 0$, and if $x_1 \leq -\epsilon$ then $\Pr_{\|\mathbf{v}\| \leq 1}\left[v_1 < -\frac{x_1}{\epsilon}\right] = 1$. Overall

$$|x_1| \geq \epsilon \implies \nabla \tilde{f}(\mathbf{x}) = \text{sign}(x_1) \cdot \mathbf{e}_1 . \tag{26}$$

Third, since probabilities are bounded between zero and one, we obtain the global upper estimate

$$\left\|\nabla \tilde{f}(\mathbf{x})\right\| \leq 1 . \tag{27}$$

Lastly, $\Pr_{\|\mathbf{v}\| \leq 1}\left[v_1 < -\frac{x_1}{\epsilon}\right]$ equals to the volume of the intersection of the halfspace $\left\{\mathbf{v} \in \mathbb{R}^d \mid v_1 < -\frac{x_1}{\epsilon}\right\}$ with the unit ball, normalized by the unit ball volume. In particular, since this intersection is a subset of the spherical sector associated with the spherical cap $\left\{\mathbf{v} \in \mathcal{S}^{d-1} \mid v_1 < -\frac{x_1}{\epsilon}\right\}$, its normalized volume is less then the surface area of the cap. By well known estimates of spherical cap (for example [4]):

$$\Pr_{\|\mathbf{v}\| \leq 1}\left[v_1 < -\frac{x_1}{\epsilon}\right] \leq \Pr_{\|\mathbf{v}\| = 1}\left[v_1 < -\frac{x_1}{\epsilon}\right] \leq \exp\left(-\frac{dx_1^2}{2\epsilon^2}\right) . \tag{28}$$

By combining Eq. (24) and Eq. (28) we get

$$\left\|\nabla \tilde{f}(\mathbf{x})\right\| \geq 1 - 2\exp\left(-\frac{dx_1^2}{2\epsilon^2}\right) .$$

In particular,

$$|x_1| \geq \frac{\sqrt{2\log(10)}\epsilon}{\sqrt{d}} \implies \left\|\nabla \tilde{f}(\mathbf{x})\right\| \geq \frac{4}{5} . \tag{29}$$

We are now ready to describe the choice of $\mathbf{x}_0$ which will prove the claim, depending on the value of $\eta$.

**Case I:** $\eta \leq \frac{5\sqrt{2\log(10)}\epsilon}{2\sqrt{d}}$

We set $\mathbf{x}_0 = \mathbf{e}_1$. First, $\mathbf{x}_0$ is indeed at distance 1 from $\mathbf{0}$, which by Eq. (25) is a stationary point. Furthermore, by the definition of gradient descent, Eq. (24) and Eq. (27), for all $t \leq \frac{2\sqrt{d}}{5\sqrt{2\log(10)}\epsilon} - \frac{2}{5}$:

$$(\mathbf{x}_{t+1})_1 = \left(\mathbf{x}_0 - \eta\left(\sum_{i=1}^t \nabla \tilde{f}(\mathbf{x}_i)\right)\right)_1$$

$$\geq 1 - \frac{5\sqrt{2\log(10)}\epsilon}{2\sqrt{d}} \cdot t \cdot 1$$

$$\geq \frac{\sqrt{2\log(10)}\epsilon}{\sqrt{d}}$$

So by Eq. (29), for every $t \leq \frac{2\sqrt{d}}{5\sqrt{2\log(10)}\epsilon} - \frac{2}{5}$: $\left\|\nabla \tilde{f}(\mathbf{x}_t)\right\| \geq \frac{4}{5}$. Consequently, the minimal $t$ for which the gradient norm is less than $\epsilon$ satisfies $t > \frac{2\sqrt{d}}{5\sqrt{2\log(10)}\epsilon} - \frac{2}{5} = \Omega(\frac{\sqrt{d}}{\epsilon})$.

**Case II:** $\frac{5\sqrt{2\log(10)}\epsilon}{2\sqrt{d}} < \eta \leq 2$

In this case, we define the real function

$$\phi(s) := 2s - \eta\left(\nabla \tilde{f}(s \cdot \mathbf{e}_1)\right)_1$$

On on hand, by assumption on $\eta$ and Eq. (29):

$$\phi\left(\frac{\sqrt{2\log(10)}\epsilon}{\sqrt{d}}\right) = \frac{2\sqrt{2\log(10)}\epsilon}{\sqrt{d}} - \eta\left(\nabla\tilde{f}\left(s\cdot\mathbf{e}_1\right)\right)_1$$

$$\leq \frac{2\sqrt{2\log(10)}\epsilon}{\sqrt{d}} - \frac{5\sqrt{2\log(10)}\epsilon}{2\sqrt{d}}\cdot\frac{4}{5}$$

$$= 0$$

On the other hand, $\frac{\eta}{2} > \frac{5\sqrt{2\log(10)}\epsilon}{4\sqrt{d}} > \frac{\sqrt{2\log(10)}\epsilon}{\sqrt{d}}$ and by Eq. (27):

$$\phi\left(\frac{\eta}{2}\right) = \eta - \eta\left(\nabla\tilde{f}\left(\frac{\eta}{2}\cdot\mathbf{e}_1\right)\right)_1$$

$$\geq \eta - \eta\cdot 1$$

$$= 0$$

Notice that $\phi$ is continuous since $\tilde{f}$ is smooth, so by the intermediate value theorem there exists $s^* \in \left[\frac{\sqrt{2\log(10)}\epsilon}{\sqrt{d}}, \frac{\eta}{2}\right]$ such that $\phi\left(s^*\right) = 0$. Equivalently,

$$s^* - \eta\left(\nabla\tilde{f}\left(s\cdot\mathbf{e}_1\right)\right)_1 = -s^* . \tag{30}$$

We set $\mathbf{x}_0 = s^*\mathbf{e}_1$. First, $\mathbf{x}_0$ is of distance at most $\frac{\eta}{2} \leq 1$ from $\mathbf{0}$, which by Eq. (25) is a stationary point. Furthermore, by the definition of gradient descent and Eq. (30) we get

$$\mathbf{x}_1 = s^*\mathbf{e}_1 - \eta\nabla\tilde{f}\left(s^*\mathbf{e}_1\right) = -s^*\mathbf{e}_1 = -\mathbf{x}_0 .$$

Inductively, due to the symmetry of $\tilde{f}$ with respect to the origin, we obtain $\mathbf{x}_t = (-1)^t\mathbf{x}_0$. In particular, since $s^* \geq \frac{\sqrt{2\log(10)}\epsilon}{\sqrt{d}}$ Eq. (29) ensures that for all $t \in \mathbb{N}$ : $\left\|\nabla\tilde{f}\left(\mathbf{x}_t\right)\right\| \geq \frac{4}{5} > \epsilon$.

**Case III:** $\eta > 2$

Set $\mathbf{x}_0 = \mathbf{e}_1$, which satisfies the distance assumption as explained in case I. By the definition of gradient descent and Eq. (26):

$$\mathbf{x}_1 = \mathbf{e}_1 - \eta\nabla\tilde{f}\left(\mathbf{e}_1\right) = (1 - \eta)\mathbf{e}_1 .$$

Notice that $(1 - \eta) < -1$, so by invoking Eq. (26) we get

$$\mathbf{x}_2 = \mathbf{x}_1 - \eta\nabla\tilde{f}\left(\mathbf{x}_1\right) = (1 - \eta)\mathbf{e}_1 + \eta\mathbf{e}_1 = \mathbf{x}_0 .$$

We deduce that for all $t \in \mathbb{N}$ : $\mathbf{x}_{t+2} = \mathbf{x}_t$, and in particular by Eq. (26): $\left\|\nabla\tilde{f}\left(\mathbf{x}_t\right)\right\| = 1 > \epsilon$. $\quad\square$

## D  Proof of Thm. 3

Our proof will closely follow the analysis employed in [24, Theorem 3] for a slightly different setting.

Fix an iteration budget $T$ and some dimension $d \geq T$. Let $A$ be the symmetric $d \times d$ tridiagonal matrix defined as

$$\forall 1 \leq i < T \qquad A(i, i) = 2 , \ A(i, i+1) = -1$$

$$\forall 1 < i \leq T \qquad A(i, i-1) = -1$$

$$A(T, T) = k := \frac{\sqrt{2} + 3}{\sqrt{2} + 1}$$

$$A(i, j) = 0 \quad \text{for all other } (i, j) .$$

Also, for some constant $b$ to be determined later, define the quadratic function

$$g(\mathbf{x}) := \mathbf{x}^\top M\mathbf{x} - \frac{1}{4}\mathbf{e}_1^\top\mathbf{x} + b \quad \text{where} \quad M = \frac{1}{8}(A + 4I) .$$

It is easily verified that this function can be equivalently written as

$$g(\mathbf{x}) = \frac{1}{8}\left(x_1^2 + \sum_{i=1}^{T-1}(x_i - x_{i+1})^2 + (k-1)x_T^2 - 2x_1\right) + \frac{1}{2}\|\mathbf{x}\|^2 + b. \tag{31}$$

We first collect a few useful facts about $g(\cdot)$, stated in the following two lemmas:

**Lemma 9.** *$M$ satisfies $\frac{1}{2} \leq \lambda_{\min}(M) \leq \lambda_{\max}(M) \leq 1$. As a result, $M$ is positive definite, and $g(\cdot)$ is strictly convex and has a unique minimum.*

*Proof.* $A$ is symmetric, and for any $\mathbf{x} \in \mathbb{R}^T$, we have

$$\mathbf{x}^\top A\mathbf{x} = x_1^2 + \sum_{i=1}^{T-1}(x_i - x_{i+1})^2 + (k-1)x_T^2.$$

This is non-negative, which establishes that $A$ is a positive semidefinite matrix. Hence, by definition of $M$, $\lambda_{\min}(M) = \frac{1}{8}(\lambda_{\min}(A) + 4) \geq \frac{1}{8} \cdot 4 = \frac{1}{2}$, which implies that $M$ is positive definite. As a result, $g(\cdot)$ is strictly convex and has a unique minimum. Also, by the displayed equation above,

$$\mathbf{x}^\top A\mathbf{x} \leq x_1^2 + 2\sum_{i=1}^{T-1}(x_i^2 + x_{i+1}^2) + (k-1)x_T^2 \leq 3x_1^2 + \sum_{i=2}^{T-1}(2x_i^2 + 2x_{i+1}^2) + (k-1)x_T^2$$

$$= 3x_1^2 + 4\sum_{i=2}^{T-1}x_i^2 + (k+1)x_T^2 \leq 4\|\mathbf{x}\|^2,$$

where we use the fact that $k \leq 3$. This establishes that $\lambda_{\max}(A) \leq 4$, and therefore $\lambda_{\max}(M) = \frac{1}{8}(\lambda_{\max}(A) + 4) \leq 1$. $\qquad\square$

**Lemma 10.** *The minimum $\mathbf{x}^*$ of $g(\cdot)$ is of the form $\mathbf{x}^* = (q, q^2, \ldots, q^T, 0, \ldots, 0)$, where $q = \frac{\sqrt{2}-1}{\sqrt{2}+1}$. Moreover, $\|\mathbf{x}^*\| \leq \sqrt{\frac{\sqrt{2}-1}{2}} < \frac{1}{2}$.*

*Proof.* By the previous lemma and the fact that $g(\cdot)$ is differentiable, $\mathbf{x}^*$ is the unique point satisfying $\nabla g(\mathbf{x}^*) = \mathbf{0}$. Thus, it is enough to verify that the formula for $\mathbf{x}^*$ stated in the lemma indeed satisfies this equation. Computing the gradient of $g(\cdot)$ using the formulation in Eq. (31)), we just need to verify that

$$6q - q^2 - 1 = 0, \quad \forall i \in \{2, \ldots, T-1\} \quad q^{i-1} - 6q^i + q^{i+1} = 0, \quad (k+4)q^T - q^{T-1} = 0,$$

or equivalently,

$$1 - 6q + q^2 = 0, \quad (k+4)q - 1 = 0,$$

which is easily verified to hold for the value of $q$ stated in the lemma. Finally, we have

$$\|\mathbf{x}^*\|^2 = \sum_{i=1}^{d}(x_i^*)^2 = \sum_{i=1}^{T}q^{2i} < \sum_{i=1}^{\infty}q^i = \frac{q}{1-q} = \frac{\sqrt{2}-1}{2},$$

implying $\|\mathbf{x}^*\| \leq \sqrt{\frac{\sqrt{2}-1}{2}}$ as required. $\qquad\square$

Finally, we assume that the constant term $b$ in Eq. (31) is fixed so that $g(\mathbf{x}^*) = 0$, which means that $g(\cdot)$ can be written in the form

$$g(\mathbf{x}) = (\mathbf{x} - \mathbf{x}^*)^\top M(\mathbf{x} - \mathbf{x}^*). \tag{32}$$

With this construction in hand, we now turn to prove the theorem. We will start with the family of zero-respecting algorithms $\mathcal{A}_{zr}$, using any dimension $d \geq T$, and take $g(\cdot)$ as the "hard" function on which we will prove a lower bound (note that by the lemmas above and Eq. (32), it satisfies the conditions stated in the theorem).

Consider any algorithm in $\mathcal{A}_{zr}$. By definition of zero-respecting algorithms, its first query point is the origin, $\mathbf{x}_1 = 0$. Now, note that by the structure of $g(\cdot)$ as specified in Eq. (31), when querying the oracle at $\mathbf{x}_1 = \mathbf{0}$, it receives a gradient supported on the first coordinate. Because of the zero-respecting assumption, it means that support$(\mathbf{x}_2) \subseteq \{1\}$, which again by Eq. (31) means that the returned gradient is supported on the first *two* coordinates. Continuing this process, it is easily seen by induction that

$$\text{support}(\mathbf{x}_t) \subseteq [t-1] \ .$$

for all $t$, and in particular, support$\{\mathbf{x}_1, \ldots, \mathbf{x}_T\} \subseteq [T-1]$. As a result,

$$\min_{t \in \{1,\ldots,T\}} \|\mathbf{x}_t - \mathbf{x}^*\|^2 \ \geq \ (x_T^*)^2,$$

which by Lemma 10 is at least $q^{2T} = \left(\frac{\sqrt{2}-1}{\sqrt{2}+1}\right)^{2T}$. Taking a square root, we get that

$$\min_{t \in \{1,\ldots,T\}} \|\mathbf{x}_t - \mathbf{x}^*\| \ \geq \ \left(\frac{\sqrt{2}-1}{\sqrt{2}+1}\right)^{T} \ \geq \ \exp(-T) \,,$$

as stated in the theorem.

We now turn to prove the theorem for deterministic algorithms. This time, we will let the dimension be any $d \geq 2T$. Fixing an algorithm, and letting $\mathbf{u}_1, \ldots, \mathbf{u}_T$ be orthonormal vectors to be specified shortly, we prove the lower bound for the function

$$\tilde{g}(\mathbf{x}) \ := \ \frac{1}{8}\left((\mathbf{u}_1^\top \mathbf{x})^2 + \sum_{i=1}^{T-1}(\mathbf{u}_i^\top \mathbf{x} - \mathbf{u}_{i+1}^\top \mathbf{x})^2 + (k-1)(\mathbf{u}_T^\top \mathbf{x})^2 - 2\mathbf{u}_1^\top \mathbf{x}\right) + \frac{1}{2}\|\mathbf{x}\|^2 + b \ . \quad (33)$$

Importantly, we note that

$$\tilde{g}(\mathbf{x}) \ = \ g(U\mathbf{x})$$

where $g(\cdot)$ is the function defined previously in Eq. (31), and $U$ is an orthogonal matrix whose first $T$ rows are $\mathbf{u}_1, \ldots, \mathbf{u}_T$, and the rest of the rows are some arbitrary completion of the first $T$ rows to an orthonormal basis. Thus, $\tilde{g}(\cdot)$ is equivalent to $g(\cdot)$ up to a rotation of the coordinate system specified by $U$. In particular, using Eq. (32), it follows that

$$\tilde{g}(\mathbf{x}) \ = \ (U\mathbf{x} - \mathbf{x}^*)^\top M(U\mathbf{x} - \mathbf{w}^*) \ = \ (\mathbf{x} - U^\top \mathbf{x}^*)^\top (U^\top M U)(\mathbf{x} - U^\top \mathbf{x}^*)$$
$$= \ (\mathbf{x} - \tilde{\mathbf{x}}^*)^\top \tilde{M}(\mathbf{x} - \tilde{\mathbf{x}}^*) \,,$$

where $\tilde{M} = U^\top M U$ and $\tilde{\mathbf{x}}^* = U^\top \mathbf{x}^*$. Thus, we see that $\tilde{g}(\cdot)$ has the form required in the theorem, with a matrix $\tilde{M}$ whose spectrum is identical to $M$, and a minimizer $\tilde{\mathbf{x}}^* = U^\top \mathbf{x}^*$ whose norm is the same as $\|\mathbf{x}^*\|$ (and therefore satisfying the conditions in the theorem).

We now specify how to choose $\mathbf{u}_1, \ldots, \mathbf{u}_T$ in the function definition, so as to get the lower bound on $\min_t \|\mathbf{x}_t - \tilde{\mathbf{x}}^*\|$: Since the algorithm is deterministic, its first query point $\mathbf{x}_1$ is known in advance. We therefore choose $\mathbf{u}_1$ to be some unit vector orthogonal to $\mathbf{x}_1$. Assuming that $\mathbf{u}_2, \mathbf{u}_3, \ldots$ are orthogonal to $\{\mathbf{u}_1, \mathbf{x}_1\}$ (which we shall justify shortly), we have by Eq. (33) that $\tilde{g}(\mathbf{x}_1)$ and $\nabla \tilde{g}(\mathbf{x}_1)$ depend only on $\mathbf{x}_1, \mathbf{u}_1$, and not on $\mathbf{u}_2, \mathbf{u}_3, \ldots$. As the algorithm is deterministic and depends only on the observed values and gradients, this means that even before choosing $\mathbf{u}_2, \mathbf{u}_3, \ldots$, we can already simulate its next iteration, and determine the next query point $\mathbf{x}_2$. We now pick $\mathbf{u}_2$ to be some unit vector orthogonal to $\mathbf{u}_1$ as well as to $\mathbf{x}_1, \mathbf{x}_2$. Again by the same considerations, if we assume $\mathbf{u}_3, \mathbf{u}_4, \ldots$ are orthogonal to $\{\mathbf{u}_i, \mathbf{x}_i\}_{i=1}^2$, we have that $\tilde{g}(\mathbf{x}_2)$ and $\nabla \tilde{g}(\mathbf{x}_2)$ depend only on $\{\mathbf{u}_i, \mathbf{x}_i\}_{i=1}^2$, and independent of $\mathbf{u}_3, \mathbf{u}_4, \ldots$. So again, we can simulate it and determine the next query point $\mathbf{x}_3$. We continue this process up to iteration $T$, where we fix $\mathbf{u}_T$ orthogonal to $\{\mathbf{u}_i, \mathbf{x}_i\}_{i=1}^{T-1}$ and to $\mathbf{x}_T$ (this process is possible as long as the dimension $d$ is at least $2(T-1) + 1 + 1 = 2T$, as we indeed assume).

As a result of this process, we get that $\mathbf{u}_T^\top \mathbf{x}_t = 0$ for all $t \in \{1, \ldots, T\}$. Also, since $\tilde{\mathbf{x}}^* = U^\top \mathbf{x}^*$, we also have $\mathbf{u}_T^\top \tilde{\mathbf{x}}^* = \mathbf{u}_T^\top U^\top \tilde{\mathbf{x}}^* = x_T^*$. Using Lemma 10, we get that for all $t \in \{1, \ldots, T\}$,

$$\|\mathbf{x}_t - \tilde{\mathbf{x}}^*\|^2 \geq (\mathbf{u}_T^\top(\mathbf{x}_t - \tilde{\mathbf{x}}^*))^2 \ = \ (0 - x_T^*)^2 = \left(\frac{\sqrt{2}-1}{\sqrt{2}+1}\right)^{2T} \,,$$

which implies that

$$\min_{t \in \{1,...,T\}} \|\mathbf{x}_t - \mathbf{x}^*\| \geq \left(\frac{\sqrt{2}-1}{\sqrt{2}+1}\right)^T \geq \exp(-T)$$

as required.

## E   Technical lemmas

**Lemma 11.** *Denote by* $f(\cdot)$ *the* $L_0$-*Lipschitz function* $\mathbf{x} \mapsto L_0|x_1|$. *Assume* $\tilde{f}(\cdot)$ *has* L-*Lipschitz gradients, and satisfies* $\left\|f - \tilde{f}\right\|_\infty \leq \epsilon$. *Then* $L \geq \frac{L_0}{8\epsilon}$.

*Proof.* Due to rescaling we can assume without loss of generality that $L_0 = 1$. Denoting by $\mathbf{e}_1$ the first standard basis vector, we have

$$\tilde{f}(-4\epsilon \cdot \mathbf{e}_1) \geq f(-4\epsilon \cdot \mathbf{e}_1) - \epsilon = 4\epsilon - \epsilon = 3\epsilon \,,$$
$$\tilde{f}(4\epsilon \cdot \mathbf{e}_1) \geq f(4\epsilon \cdot \mathbf{e}_1) - \epsilon = 4\epsilon - \epsilon = 3\epsilon \,,$$
$$\tilde{f}(\mathbf{0}) \leq f(\mathbf{0}) + \epsilon = \epsilon \,.$$

By the mean value theorem, there exist $-4\epsilon < t_0 < 0$, $0 < t_1 < 4\epsilon$ such that

$$\frac{\partial}{\partial x_1}\tilde{f}(t_0) = \frac{\tilde{f}(\mathbf{0}) - \tilde{f}(-4\epsilon \cdot \mathbf{e}_1)}{4\epsilon} \leq \frac{\epsilon - 3\epsilon}{4\epsilon} = -\frac{1}{2} \,,$$
$$\frac{\partial}{\partial x_1}\tilde{f}(t_1) = \frac{\tilde{f}(4\epsilon \cdot \mathbf{e}_1) - \tilde{f}(\mathbf{0})}{4\epsilon} \geq \frac{3\epsilon - \epsilon}{4\epsilon} = \frac{1}{2} \,.$$

So by Cauchy-Schwarz and $L$-smoothness of $\tilde{f}$:

$$1 = \left|\frac{\partial}{\partial x_1}\tilde{f}(t_1) - \frac{\partial}{\partial x_1}\tilde{f}(t_0)\right| = \left|\left\langle \nabla\tilde{f}(t_1 \cdot \mathbf{e}_1) - \nabla\tilde{f}(t_0 \cdot \mathbf{e}_1), \mathbf{e}_1\right\rangle\right|$$
$$\leq \left\|\nabla\tilde{f}(t_1 \cdot \mathbf{e}_1) - \nabla\tilde{f}(t_0 \cdot \mathbf{e}_1)\right\| \leq L|t_1 - t_0| \leq L \cdot 8\epsilon$$

$\square$

**Lemma 12.** *If* $\tilde{f}(\cdot)$ *has* L-*Lipschitz gradients and satisfies* $\|f - \tilde{f}\|_\infty \leq \epsilon$ *for some 1-Lipschitz function* $f(\cdot)$, *then for all* $\mathbf{x} \in \mathbb{R}^d$ : $\|\nabla\tilde{f}(\mathbf{x})\| \leq 1 + 2\epsilon + \frac{L}{2}$.

*Proof.* Let $\mathbf{x}, \mathbf{y} \in \mathbb{R}^d$. Denote $\gamma(t) := (1-t) \cdot \mathbf{x} + t \cdot \mathbf{y}$, and notice that

$$\tilde{f}(\mathbf{y}) - \tilde{f}(\mathbf{x}) = \tilde{f}(\gamma(1)) - \tilde{f}(\gamma(0)) = \int_0^1 \left(\tilde{f} \circ \gamma\right)'(t)\,dt = \int_0^1 \left\langle \nabla\tilde{f}(\gamma(t)), \gamma'(t)\right\rangle dt$$
$$= \int_0^1 \left\langle \nabla\tilde{f}(\gamma(t)), \mathbf{y} - \mathbf{x}\right\rangle dt \,. \tag{34}$$

Combining Cauchy-Schwarz with the fact that $\nabla\tilde{f}$ is $L$-Lipschitz, we get

$$\left\langle \nabla\tilde{f}(\mathbf{x}) - \nabla\tilde{f}(\gamma(t)), \mathbf{y} - \mathbf{x}\right\rangle \leq \left\|\nabla\tilde{f}(\mathbf{x}) - \nabla\tilde{f}(\gamma(t))\right\| \cdot \|\mathbf{y} - \mathbf{x}\| \leq L\|\mathbf{x} - \gamma(t)\| \cdot \|\mathbf{y} - \mathbf{x}\|$$
$$\implies \left\langle \nabla\tilde{f}(\gamma(t)), \mathbf{y} - \mathbf{x}\right\rangle \geq \left\langle \nabla\tilde{f}(\mathbf{x}), \mathbf{y} - \mathbf{x}\right\rangle - L\|\gamma(t) - \mathbf{x}\| \cdot \|\mathbf{y} - \mathbf{x}\|$$

Plugging this into Eq. (34) gives

$$\tilde{f}(\mathbf{y}) - \tilde{f}(\mathbf{x}) \geq \int_0^1 \left( \left\langle \nabla \tilde{f}(\mathbf{x}), \mathbf{y} - \mathbf{x} \right\rangle - L \|\gamma(t) - \mathbf{x}\| \cdot \|\mathbf{y} - \mathbf{x}\| \right) dt$$

$$= \left\langle \nabla \tilde{f}(\mathbf{x}), \mathbf{y} - \mathbf{x} \right\rangle - L \|\mathbf{y} - \mathbf{x}\| \cdot \int_0^1 \|\gamma(t) - \mathbf{x}\| \, dt$$

$$= \left\langle \nabla \tilde{f}(\mathbf{x}), \mathbf{y} - \mathbf{x} \right\rangle - L \|\mathbf{y} - \mathbf{x}\| \cdot \left[ \frac{1}{2} \|\gamma(1) - \mathbf{x}\|^2 - \frac{1}{2} \|\gamma(0) - \mathbf{x}\|^2 \right]$$

$$= \left\langle \nabla \tilde{f}(\mathbf{x}), \mathbf{y} - \mathbf{x} \right\rangle - \frac{L}{2} \|\mathbf{y} - \mathbf{x}\|^3$$

$$\implies \left\langle \nabla \tilde{f}(\mathbf{x}), \mathbf{y} - \mathbf{x} \right\rangle \leq \tilde{f}(\mathbf{y}) - \tilde{f}(\mathbf{x}) + \frac{L}{2} \|\mathbf{y} - \mathbf{x}\|^3 \ .$$

We assume $\|\nabla \tilde{f}(\mathbf{x})\| \neq 0$ since otherwise the desired claim is trivial. In particular, if $\mathbf{y} = \mathbf{x} + \frac{\nabla \tilde{f}(\mathbf{x})}{\|\nabla \tilde{f}(\mathbf{x})\|}$ then $\|\mathbf{y} - \mathbf{x}\| = 1$ and inequality above reveals

$$\left\| \nabla \tilde{f}(\mathbf{x}) \right\| \leq \tilde{f}(\mathbf{y}) - \tilde{f}(\mathbf{x}) + \frac{L}{2} \leq f(\mathbf{y}) - f(\mathbf{x}) + 2\epsilon + \frac{L}{2} \leq \|\mathbf{y} - \mathbf{x}\| + 2\epsilon + \frac{L}{2} = 1 + 2\epsilon + \frac{L}{2} \ ,$$

where we used the fact that $\|\tilde{f} - f\|_\infty \leq \epsilon$, and that $f$ is 1-Lipschitz. $\qquad \square$