# OpenReview forum: "Oracle Complexity in Nonsmooth Nonconvex Optimization"
_NeurIPS.cc/2021/Conference — NeurIPS 2021 Oral_

### Official Review · Reviewer_KNSx · 2021-07-09

**Rating:** 9
**Confidence:** 4

**Summary:**

This paper investigates the question of finding near $\epsilon$-stationary points for
Lipschitz nonconvex functions. This is a quite challenging question, and the paper addresses it with the following advances:

1. An oracle lower bound for the task of finding near $\epsilon$-stationary points. This is achieved by an intrincate construction that resembles the trick of hiding a cone in a random direction. This result is restricted to deterministic and zero-respecting algorithms.

2. The second contribution is the study of smoothings from an oracle perspective. It is shown in particular that in this oracle model, the Lipschitz gradient bounds obtained by randomized smoothing are essentially optimal, which rules out e.g. efficient implementations of the Lasry-Lions regularization. BTW, I wonder if Corollary 3 readily implies that computing saddle points has exponential oracle complexity with respect to the dimension (these type of results are already known, but this would be a more transparent proof, I believe).

3. A less emphasized third contribution is the observation that the relaxation used in Zhang et al. 2020 (based on the convex hull of a union of subdifferentials around the point) does not lead to anything resembling a nearly stationary point.


**Limitations And Societal Impact:**

Yes

**Main Review:**

This paper is a great addition to the understanding of nonconvex optimization. It provides
a reasonably sized proof that finding nearly $\epsilon$-stationary points is difficult, it provides a nice model to argue about smoothings algorithmically, and touches upon other works addressing similar questions. This material is clearly relevant for the NeurIPS community.

In terms of the cons, it is a problem the fact that the oracle lower bound for nearly $\epsilon$-stationary points is restricted to deterministic zero-respecting algorithms, though I share the authors view that this might be an artifact of the current analysis, and it could be resolved in follow up work. On a related note, regarding remark 5: could you use a black box function which enjoys the property of Them 3 without being a quadratic (e.g., the one in Woodworth and Srebro)? Perhaps this is an easier way to extend the applicability of the result without having to deal with the lower bound for general algorithms on quadratics, which is known to be difficult.

**Time Spent Reviewing:**

4

---

> ### Author Response · Authors · 2021-08-08
> **Review KNSx response**
>
>  We thank the reviewer for his comments.
> In fact, we are currently working on an improved version of this result, which applies to any (possibly randomized) algorithm and does not rely on a quadratic construction. However, the proof construction is subtle and rather different than the current one, and will only appear in future work (of course, we ask the reviewer to evaluate this paper based on the current results).

---

> > ### Comment · Reviewer_KNSx · 2021-08-13
> > **Answer to authors' response**
> >
> > Thank you. Just to clarify: I am not asking for direct extensions to be addressed in this round of reviews.

---

### Official Review · Reviewer_Hx4f · 2021-07-13

**Rating:** 8
**Confidence:** 4

**Summary:**

The authors consider the problem of nonsmooth nonconvex optimization through an oracle complexity perspective.

The authors propose a precise mathematical statement of the idea that there is no complexity estimate for nonsmooth nonconvex optimization based on first order oracles. This is the first contribution presented in the manuscript. The convergence criterion is a quantitative version of the property of "being close to an almost critical point". The authors show that obtaining such an approximate critical point is not possible in a dimension independant finite time, in general.

As a second contribution, the authors consider the possibility to use smoothing techniques for nonsmooth nonconvex optimization. For a smoothed objective, usual optimality criteria, such as small gradient norm, can be used. Yet classical first order methods come with complexity estimates which depend on the Lipschitz constant of the gradient of the objective. It is a known fact that randomized smoothing (formally, convolution with a density) allows to obtain Lischitz constants which have a square root dependency in the dimension. The authors show that this is actually tight in the sense that a large class of computationally implementable local smoothers need to account for such a dimension dependency.




**Limitations And Societal Impact:**

Yes

**Main Review:**

The proposed analysis is well presented and correct up to my understanding. I found the manuscript very clear and well written, the arguments and proofs are carefully detailed and the written material is of very high quality. The questions adressed by this paper are timely and the proposed answers are very satisfactory. I believe that this is of relevance to the machine learning community for which nonsmooth nonconvex optimization are central.


- About semi-algebraicity (and regularity).
The proposed counter-examples are actually semi-algebraic. The oracle complexity perspective is usually presented in a convex optimization context, which enforces a lot of structure. In the nonconvex world, there is a very large gap between general Lipschitz functions and more structured functions (such as semi-algebraic). I believe that it is worth emphasizing that the proposed counter examples are semi-algebraic. This is stronger than most classes cited in Remark 1 as Lipschitz semi-algebraic functions are both Hadamard semi-differentiable and Whitney stratifiable.

The main mechanism used in [15] for "stratifiable functions" is a projection formula which was introduced by the following reference, which is worth citing:
Bolte, J., Daniilidis, A., Lewis, A., & Shiota, M. (2007). Clarke subgradients of stratifiable functions. SIAM Journal on Optimization, 18(2), 556-572.

I do not believe that the reference to regular functions in Remark 1 is appropriate. Regularity means that all non-smoothness are qualitatively similar to convex like nonsmoothness. For example the function t -> -max(0,t) is not regular. I am not sure that the constructed counter examples are regular.

- Relation between smoothing and near-approximately-stationary.
It would be interesting to have a more detailed discussion of the relation between the two contributions. Although it is not explicitely stated, it is implicit in the second contribution that the overall goal is to find a near critical point of a smoothed version of the objective. The authors prove that this goal cannot be achieved in a dimension independant way. However, I think that being a near critical point of a smoothed version of the objective may suffer from the same drawback as described in appendix B: creation of spurious critical points which are far from any "real" critical point. At least for local averaging, I strongly believe that a similar phenomenon as in appendix B could be described: smoothing creates an artificial critical point in an area where the original function has none. Since the targets of the two contributions is overall different, it would be interesting to emphasize and discuss these differences between near criticalilty for a smoothed objective and the proposed near-approximate-stationarity.


- Relevance of qualitative / assymptotic results.
The manuscript starts by emphasizing limitation of qualitative results which are of purely asymptotic nature. The authors then proceed to show that the proposed notion of near-approximate-stationarity is not accessible in a dimension independent way and conclude that one should resort to more relaxed notions of approximate criticality (possibly producing spurious unexpected artifacts).

I think that in addition it should be explicitely stated and emphasized that the asymptotic analysis, presented in [8,15] for example, do prove that near-approximately-stationary points will be reached in finite time (without explicit estimates). The authors show that this goal is out of reach if we impose this finite time not to depend on the dimension of the problem or a specific initialisation. To me these highlight the relevance of asymptotic / qualitative results in this context as the author essentially show that there is no dimension independent quantitative counterpart. Such results remain the only available way to prove that even the simplest subgradient method has a minimizing behavior and is attracted by critical points in nonsmooth nonconvex optimization.



**Time Spent Reviewing:**

10

---

> ### Author Response · Authors · 2021-08-08
> **Review Hx4f response**
>
> Thank you very much for your detailed and helpful comments, we will incorporate them into the revised version of the paper.

---

### Official Review · Reviewer_nyBG · 2021-07-17

**Rating:** 8
**Confidence:** 3

**Summary:**

### Update

I'd like to thank the authors for their response.

I have decided to increase my score to 9 in light of the other reviews and these comments. I look forward to seeing this work published.

========

This submission investigates two problems central to non-smooth, non-convex optimization: i) the oracle complexity of computing approximately stationary points and ii) the complexity of smoothing non-convex functions.
To tackle the first problem, the authors propose a new concept of _near approximately-stationary_ points, which the set of points within $\delta$ (in some metric) of an $\epsilon$ stationary point.
This differs from previous work [35], which classified $(\epsilon, \delta)$-stationarity as $\epsilon$-stationarity of a convex combination of (generalized) gradients in a local neighbourhood of size $\delta$.
The authors prove that, unlike for $(\epsilon, \delta)$-stationarity, finding near approximately-stationary points is impossible in finite time for the classes of deterministic and randomized, zero-respecting algorithms under standard assumptions and for sufficiently small $\epsilon, \delta$.
For the second problem, the submission formalizes trade-offs between the degree of smoothness in the smooth approximation, as measured in the Lipschitz modulus of the gradient, and the computational complexity of the smoothing operation.
In particular, it is shown that for every smoothing algorithm which has polynomial running-time in the problem parameters, there exists a function for which the modulus of the smoothed function is $\tilde O(\sqrt{d})$.
As a consequence, randomized smoothing by convolution with a Gaussian or uniform distribution is seen to obtain the optimal dimension dependence out of polynomial-time smoothing procedures.

**Ethical Concerns:**

None.

**Limitations And Societal Impact:**

Yes. This is a theoretical paper with no forseeable societal impact.

**Main Review:**

# Detailed Comments

This work is a good submission with two significant theoretical contributions.

Computing stationary points of non-smooth, non-convex functions is a central task in current machine learning practice due to the ubiquity of deep neural networks with ReLU activations.
The strong impossibility result for finding such finding critical points that the authors present is both novel and of practical importance to optimizers in the NeurIPS community.

The theory on smoothing operators for non-smooth, non-convex functions is also interesting, but less immediately attractive in the context of neural networks;
smooth activation functions can always be used to obtain differentiable, if not Lipschitz smooth, networks and I am not aware of an alternative class of non-convex, non-smooth problems where smoothing is an attractive idea.
With that said, the result is new and it has interesting consequences for the optimality of different smoothing operators, as the authors point out.

The major weakness of the submission is the limited connection between the two theoretical results.
The two major theorems are largely related by their shared setting in non-convex, non-smooth optimization and at times the paper reads like two separate submissions.
Indeed, I feel that both results might work better and command more interest if further developed in separate works: one devoted to finding stationary points and one devoted to smoothing procedures.

For example, an interesting questing arising from the first result is the following:
what kind of reasonable guarantees can be expected for subgradient methods in the non-convex setting?
This question is especially interesting since the authors show that $(\epsilon, \delta)$-stationarity is unsuitable.

## Writing

The writing in the main paper is very polished.
However, I found the proof of Theorem 2 in Appendix A.2 quite difficult to follow.
Perhaps this section and section A.1 could be improved by the inclusion of a "roadmap" outlining of the argument.

I think Figure 1 is excellent and really contributes to an intuitive understanding of the construction for Theorem 1.


## Theory

I checked the proofs, which appear to be correct. I have several small comments/questions.

- Line 708 (18): How can Lemma 11 be applied here to control the value of $L$? So far we have made no assumptions on the form of $f$, while Lemma 11 assumes $f(\mathbf{x}) = L_0 |x_1|$.
The conclusion derived from Lemma 11, namely,
$$
\sqrt{\log((M + 1))\log((T + 1))} \leq \frac{\sqrt{d}}{32r}
$$
is necessary for $\Delta$ to be a grid on the unit interval.
Can the authors please clarify this issue?

- Display after line 908 (page 28): Shouldn't it be $q^{2i}$, rather than $q^i$? The subsequent bound still holds, of course.

- Line 652 (page 16): Shouldn't this be "between $\|x - x^* \|$ and $\frac{1}{\sqrt{2}}\|x - x^*\|$"? Well, I suppose that this is a slightly tighter bound and the bound $\lambda_{\text{min}}(M^{1/2}) > 1 / 2$ also holds.

- Footnote 7 (page 17): Should the last line read "In particular, this bad event will occur with probability larger than $T \exp\{-d/18\}$ for some realization of the \*\*algorithm's randomness\*\*"?
    Otherwise I do not see how the desired contradiction arises.

- Eq. 10 (page 18): I believe the left-hand side of the equation is missing a factor of $2$.

## Experiments

N/A

## Questions for the Authors

- The set of "zero-respecting" algorithms does not include many simple and interesting randomized algorithms.
For example:
 i) choose a random point $z \in \mathbb{R}^d$,

 ii) take $x_k^* =$ argmin  $\{ f(y) : y \in \{ z, x_k } \} $,

 iii) set
$x_{k+1} = x_k^+ - \eta_k \nabla f(x_k^+)$.


I really don't see what value the zero-respecting assumption has has over the classical "remains in the span of the gradients" assumption aside from being technically weaker.
Can the authors suggest an interesting algorithm which falls in one class but not the other?

- The impossibility result for finding stationary points relies on a dimension-dependent argument where $d \geq 2 T$ is necessary.
Do the authors think that a dimension-independent impossibility result is feasible?

**Time Spent Reviewing:**

9

---

> ### Author Response · Authors · 2021-08-08
> **Review nyBG response**
>
> We thank the reviewer for his comments.
> - We completely agree that finding the "right" finite-time performance metric for subgradient methods in the nonconvex setting is an interesting open question (which we tried to raise in the Discussion section).
> - We will add clarifications in Appendix A to clarify the proofs.
> - Line 708 comment: Lemma 11 implies that for \epsilon<1, there exists a 1-Lipschitz function (x\mapsto |x_1|), which is the function class over which the smoother is defined, that cannot be \epsilon approximated with an L-smooth function for any L<1/8. Therefore, if \epsilon<1, then an L-smoother exists in the first place only when L\geq1/8. Indeed as the reviewer pointed out, we use this observation as a reduction, in order for our construction to be well defined on the unit interval. We will make an effort to clarify this.
> - Display after line 908 + line 651 + Eq. 10 + Footnote 7 - indeed, thank you for catching these.
> - An example for a natural class of algorithms which are zero-respecting but not span-respecting is coordinate descent methods (e.g. "Efficiency of Coordinate Descent Methods on Huge-Scale Optimization Problems" by Nesterov). In any case, we agree that linear-span algorithms already contain most of the algorithms of interest, but our proof easily extends to this larger class, so we saw no reason not to present it in that generality.
> - Some assumption on the dimension d being large enough is necessary: When d=1, finding an approximately-stationary point can be easily achieved via binary search. The same issue also occurs in other standard oracle complexity bounds in the literature.

---

> > ### Comment · Reviewer_nyBG · 2021-08-08
> > **Thanks for Clarifying Comments**
> >
> > I appreciate the authors comments.
> >
> > - Many thanks for addressing the issue in Line 708; I follow the argument now. I think including essentially the same comments in the appendix should be sufficient to clarify the proof.
> >
> > - Right, coordinate-descent methods are a great example that don't respect the gradient span. A one-sentence comment to this effect in the main paper would be a nice addition.
> >
> > - Of course, Lipschitz optimization when $d=1$ is an "easy" problem. Closer to my original thought is to fix $d > 1$ and ask for what $\epsilon$-accuracy does finding a stationary point have a similar cost (i.e. exponential in $d$) to Lipschitz optimization over a compact set.

---

### Official Review · Reviewer_SE7Y · 2021-07-17

**Rating:** 7
**Confidence:** 4

**Summary:**

This is a theoretical paper on the oracle complexity of nonconvex and nonsmooth optimization problem. The paper mainly consists of two results: one is a negative result showing that it is impossible to get a near-approximately stationary points with worst-case finite-time guarantees for a large class of optimization algorithms; the other one is a positive result showing the trade-off between the computational efficiency and smoothness of the approximation scheme indeed exists.

**Limitations And Societal Impact:**

This is a purely theoretical paper on the complexity of numerical algorithms. I felt this work does not have negative societal impact.

**Main Review:**

Although the literature on nonconvex and nonsmooth optimization problem is overwhelming in recent years, I felt the current manuscript adds value to this crowded line of research. In particular, it demonstrates the difficulty for getting an approximate stationary point in the simultaneous presence of the nonconvexity and nonsmoothness from the complexity perspective. The paper is written clearly and very easy to follow. In particular, the constructive proof of Theorem 1 is nice and generalizable.

One major question that I have for this paper is on the definition of a near-approximately stationary solutions. My feeling is that the epsilon-stationary point is not defined very well here. Consider the example that f(x) = \|x\|_1. Obviously x=0 is the only stationary point. However, if one defines the epsilon-approximate stationary point as the existence of u\in \partial f(x) such that \|u\| \leq epsilon, then this set of approximate stationary solutions is still a singleton \{x=0\}, as long as epsilon <1. This means that even for such a simple convex function, the ``approximate'' solution may be as strong as an exact stationary solution, which is of course not desirable. In fact, the definition of approximate stationary solutions in this work is equivalent to \{x : distance(0, \partial f(x)) \leq epsilon\}. Since \partial f is a set-valued mapping that is potentially discontinuous due to the nonsmoothness of f, this definition is not robust and the above mentioned issue is likely to happen. Perhaps a more relaxed definition of approximate solution may lead to a better worst case result.

Another question is on the assumption of the global Lipschitz continuity of the function f. This assumption is quite strong and in particularly excludes the deep neural network problems with more than one layer. However, Rademacher's theorem only requires the local Lipschitz continuity of the function. This means both the definition of (approximate) stationarity and the randomized smoothing algorithm can be applied directly with the much weaker local Lipschitz continuity. Can the authors explain the necessity to assume the global Lipschitz continuity throughout the paper?

**Time Spent Reviewing:**

3

---

> ### Author Response · Authors · 2021-08-08
> **Review SE7Y response**
>
> We thank the reviewer for his comments.
> - We completely agree that a more relaxed definition of approximate stationarity may lead to better worst-case results -- indeed, finding such a definition is one of the open questions we raise in the Discussion section.
> - Since the paper provides hardness results, assuming the function is globally Lipschitz only strengthens the result. In other words, since our hardness result already applies to globally Lipschitz functions, the same result applies to any larger function class which contains it (e.g. locally Lipschitz functions, as suggested by the reviewer.

---

### Official Review · Reviewer_vcVQ · 2021-07-20

**Rating:** 8
**Confidence:** 3

**Summary:**

This paper presented strong technical results for nonsmooth nonconvex problems. The main lower bound is counter-intuitive; the argument for random smoothing is deep. Meanwhile, the main body gives a very good illustration of the hard technical content.

**Limitations And Societal Impact:**

No Societal Impact

**Main Review:**

I like this paper as I learned a lot about nonsmooth nonconvex optimization from it. Many contents are new for me such as generalized gradients and Lasry-Lions Regularization, but the writing makes them accessible. The content about random smoothing is surprising but reasonable.

Some minor comments are that I do not find the exact $(\delta, \epsilon)$ Near Approximately-Stationary Points. Maybe it is just because the notation is changed in Theorem 1.

Some open question is that despite the theoretical meaning, are there any implications of this work for practical problems? It seems that most of the problems we encounter are structured. Even for deep neural networks, we can still consider smoothing ReLu by smoothing nonlinearity. I think that such a strong lower bound may be not application for most of the nonsmooth nonconvex problems we need to solve.



**Time Spent Reviewing:**

5

---

> ### Author Response · Authors · 2021-08-08
> **Review vcVQ response**
>
> We thank the reviewer for his comments.
> -  The definition of a (\delta,\epsilon) stationary point appears in Appendix B, where we discuss it. We chose not to provide it in the main paper, since it is not the focus of our work.
> - Our work indeed does not exclude finding approximately stationary points for some specific neural network architecture. However, the lower bound construction can tell us what assumptions might be needed in order to circumvent the lower bound and obtain a positive result.

---

### Decision · Program_Chairs · 2021-09-27

**Decision:**

Accept (Oral)

**Comment:**

This paper provides  oracle complexity of nonconvex and nonsmooth optimization problem. The paper provides useful and interesting results about the complexity of getting a near-approximated stationary points, and the techniques and results are new to the literature.

Overall the paper is very well-written, easy to follow, but the results are substantial. I recommend acceptance of this paper at least as a spotlight paper.